# Newly deceased Caribbean reef-building corals experience rapid carbonate loss and colonization by endolithic organisms

Francisco Medellín-Maldonado [1,2,3✉], Israel Cruz-Ortega[4], Esmeralda Pérez-Cervantes [1], Orion Norzogaray-López[5], Juan P. Carricart-Ganivet[4], Andrés López-Pérez [3] & Lorenzo Alvarez-Filip [2]

Coral mortality triggers the loss of carbonates fixed within coral skeletons, compromising the reef matrix. Here, we estimate rates of carbonate loss in newly deceased colonies of four Caribbean reef-building corals. We use samples from living and recently deceased colonies following a stony coral tissue loss disease (SCTLD) outbreak. Optical densitometry and porosity analyses reveal a loss of up to 40% of the calcium carbonate ($CaCO_3$) content in dead colonies. The metabolic activity of the endolithic organisms colonizing the dead skeletons is likely partially responsible for the observed dissolution. To test for the consequences of mass mortality events over larger spatial scales, we integrate our estimates of carbonate loss with field data of the composition and size structure of coral communities. The dissolution rate depends on the relative abundance of coral species and the structural properties of their skeletons, yet we estimate an average reduction of 1.33 kg $CaCO_3$ m$^{-2}$, nearly 7% of the total amount of $CaCO_3$ sequestered in the entire system. Our findings highlight the importance of including biological and chemical processes of $CaCO_3$ dissolution in reef carbonate budgets, particularly as the impacts of global warming, ocean acidification, and disease likely enhance dissolution processes.

[1] Posgrado en Ciencias del Mar y Limnología, Universidad Nacional Autónoma de México, Av. Ciudad Universitaria 3000, Coyoacán, 04510 Ciudad de México, Mexico. [2] Biodiversity and Reef Conservation Laboratory, Unidad Académica de Sistemas Arrecifales en Puerto Morelos, ICML, UNAM, 77580 Puerto Morelos, Mexico. [3] Laboratorio de Arrecifes y Biodiversidad, Universidad Autónoma Metropolitana, 09340 Ciudad de México, Mexico. [4] Laboratorio de Esclerocronología de Corales Arrecifales, Unidad Académica de Sistemas Arrecifales en Puerto Morelos, ICML, UNAM, 77580 Puerto Morelos, Mexico. [5] Instituto de Investigaciones Oceanológicas y Facultad de Ciencias Marinas, Universidad Autónoma de Baja California, 22860 Ensenada, BC, Mexico. ✉email: frame@comunidad.unam.mx

Coral reefs are key sites for biodiversity and provide crucial ecosystem services[1]. The three-dimensional structure of each coral reef is a product of $CaCO_3$ production and erosion stemming from the environment and a multitude of organisms, with these processes acting across various spatiotemporal scales[2–4]. For coral reefs to grow, calcification must exceed the loss of $CaCO_3$ due to physical, chemical, and biological erosion[4,5]. However, the delicate balance between production and erosion has been severely altered in many reefs worldwide due to human-induced changes in reef ecology and the environment[5]. This imbalance has led to a decline in the production of $CaCO_3$ by corals and an increase in carbonate removal due to destructive processes, which has caused reefs to enter states of low carbonate production or even net erosion[6]. A transition to erosive states is particularly concerning because of the concomitant severe modifications to reef frameworks and coral growth that compromise reef functioning[7]. This transition is even more concerning when modeled scenarios predict that environmental conditions will promote erosive processes and thus further limit coral growth[5,8,9].

Our understanding of the ways in which rapid changes in reef ecology affect reef carbonate dynamics centers on construction processes; the abilities of reef-building corals to secrete $CaCO_3$ under conditions of elevated temperature, acidification, and nutrient concentrations; and the implications of coral mortality in terms of the potential reduction in carbonate production (e.g. refs. [8,10,11]). In spite of what we have learned, the erosive processes that follow coral mortality are far less understood[4,12,13] which largely reflects our lack of knowledge of the activities of organisms, such as grazers (e.g., parrotfish and sea urchins) and macroborers (e.g., sponges, molluscs and polychaetes), that erode the surfaces and interiors of living and dead coral skeletons[14,15]. However, the consequences of dissolution, especially those that follow coral mortality, are poorly understood[4,16] as most are complex and involve multiple taxa and environmental factors[4,17,18]. Skeletal dissolution can be driven by microboring organisms, such as algae and bacteria, or it may be chemical and caused by low pH levels in ocean water. Hereafter we refer to these two dissolution processes simply as *net dissolution*.

After a mortality event, naked skeletons are subject to two processes that increase net dissolution. First, the dead skeleton surface is rapidly covered by biofilms that also colonize its interior[19,20]. Second, existing microbial endolithic communities (e.g., bacterial and filamentous algae assemblages) bloom due to the resulting increase in light that is able to penetrate the bleached skeleton[21–23]. These two processes drive other well-known biogeochemical processes that produce $CO_2$, such as organic matter descomposition and respiration, which decrease the saturation state of carbonate minerals (i.e., aragonite) and reduce pH, thus leading to carbonates dissolution[20,23,24]. The net dissolution driven by these processes can result in losses of up to 1.1 kg $CaCO_3$ m$^{-2}$ year$^{-1}$ [16,25,26]. At the same time, increases in the bacterial and algal communities that colonize dead skeletons likely affect the balance between photosynthesis and respiration, and between calcification and dissolution, thus favoring erosive processes, particularly those that occur at night[27–29]. The interstitial water that penetrates the pores of coral skeletons devoid of living tissue takes on pH values that are up to 0.8 units lower than the pH values of the surrounding water due to the metabolic activity of endolithic communities[24,30]. Together, these processes create corrosive conditions that favor carbonate dissolution, which reduces skeletal density and increases skeletal porosity[9,24,31]. $CaCO_3$ dissolution is particularly concerning, as the combined effect of acidification and rising temperatures is known to accelerate net dissolution within dead coral skeletons[19,32]. Ultimately, when the interiors of coral skeletons are weakened, coral structures become more susceptible to other sources of bioerosion and mechanical damage like wave action[16,33].

Internal macrobioerosion and the net dissolution of coral skeletons are particularly concerning during mass mortality events, such as those due to widespread bleaching or disease outbreaks that result in the loss of the protective cover of coral tissue[18,32]. Following these events, large quantities of coral skeletons become susceptible to destructive processes, which likely results in drastic changes in gross $CaCO_3$ production (see refs. [34–36]). However, measuring net dissolution is difficult and has only been successful in controlled environments or on small spatial scales[19,24,32,37] which limits our understanding of the consequences of mass mortality events at the reef scale.

In this study, we measured density changes produced by net dissolution to quantify $CaCO_3$ mass loss in the dead skeletons of four reef-building corals after an outbreak of stony coral tissue loss disease (SCTLD) in the Puerto Morelos reef system[38]. We used densitometry and porosity analyses to investigate changes in skeletal density in colonies of *Dendrogyra cylindrus*, *Siderastrea siderea*, *Pseudodiploria strigosa*, and *Orbicella faveolata* before and after they died and related these changes to species growth, density, and skeletal architecture. Our study is framed within an SCTLD outbreak, and thus we also used colony-level estimates of carbonate loss to quantify the potential consequences of net dissolution at the reef scale. Ultimately, our results demonstrate that carbonate loss within dead coral skeletons is severe, even in non-acidified environments.

## Results

**Changes in skeletal density and porosity.** We measured changes in skeletal density and porosity before and after mortality in two ways (Fig. 1a). For *D. cylindrus*, we obtained core samples from colonies in 2015, when they were alive, and then sampled the same colonies in 2019 and 2020, 1 and 2 years after they had died in 2018 (Table 1A). For *S. siderea*, *P. strigosa*, and *O. faveolata*, we obtained samples from living colonies during the sampling campaign of 2020, during which we also sampled colonies that had died due to the SCTLD outbreak in 2019 (see "Methods"; Table 1B). Given the differences in the skeletal structure of each species and collected samples, we used the density bands (i.e., multiple measures per core; Supplementary Table 1) to create separate linear mixed models (LMM) for each species to test for changes in skeletal density while accounting for the effects of colony identity, colony status (living, dead 1 year, dead 2 year), annual growth, and reef zone (see "Methods"; Fig. 2a–h).

We found strong evidence of density loss in carbonate structures after coral tissue mortality for three out of the four studied species (Fig. 2 and Supplementary Table 2). Separate LMM indicated that skeletal density was significantly lower after mortality in *D. cylindrus* ($\chi^2 = 14.07$, df = 2, $p < 0.01$), *P. strigosa* ($\chi^2 = 10.86$, df = 1, $p < 0.01$), and *S. siderea* ($\chi^2 = 6.02$, df = 1, $p = 0.01$). While we did not observe significant differences between the skeletal density of dead and living colonies of *O. faveolata* ($\chi^2 = 0.009$, df = 1, $p = 0.92$; Fig. 2h), the porosity analyses supported the findings obtained by optical densitometry. Higher porosity was observed in dead *D. cylindrus*, *S. siderea*, and *P. strigosa* colonies when compared to that of living colonies (Supplementary Fig. 1).

For *D. cylindrus*, we were also able to explore the progression of skeletal density loss using data from the two sampling periods following the mortality event. We found significant differences in the colony density before and after the colonies died (1 year after death, $z = 10.798$, $p < 0.01$; 2 years after death, $z = 6.586$, $p < 0.01$). However, no differences in density were found between colonies 1 year after death and 2 years after death ($z = -2.27$, $p <= 0.06$; Fig. 2e). This lack of differences suggests that the most pronounced change occurred during the first year following mortality.

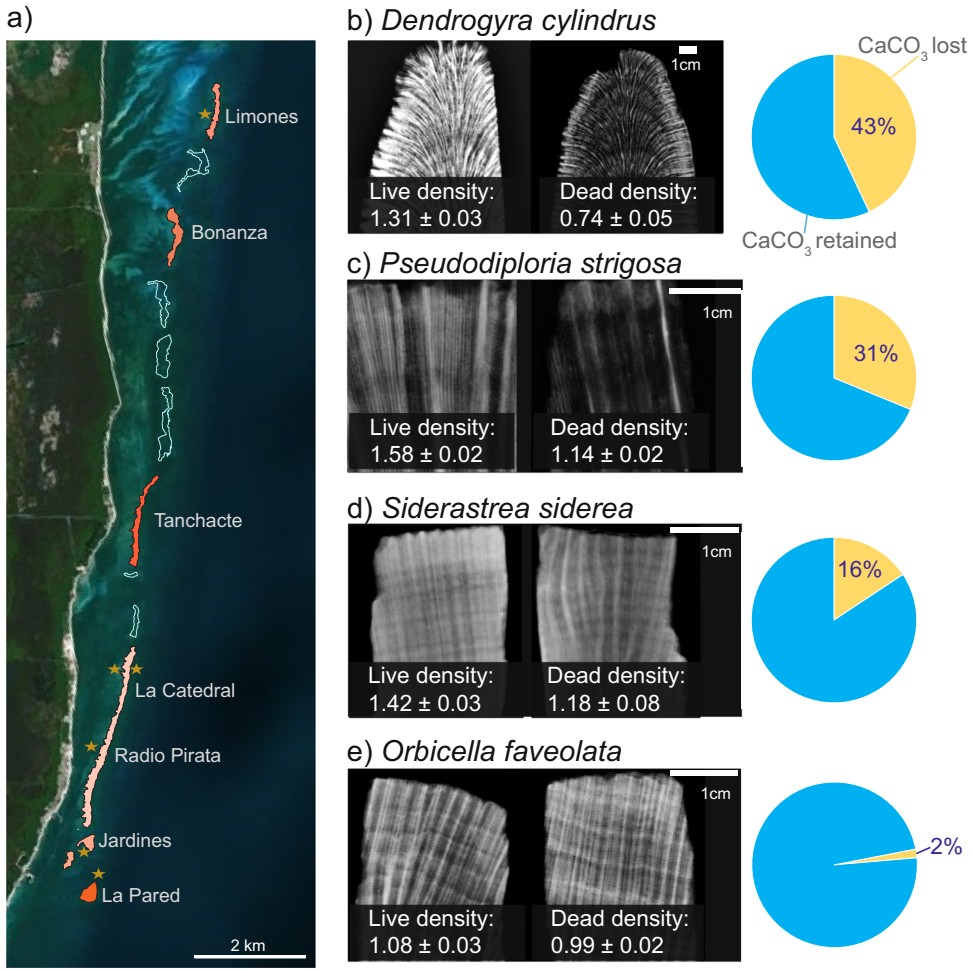

**Fig. 1 Reefs surveyed in the Puerto Morelos reef system and density loss in live and dead coral colonies. a** Polygons of the reefs and data used to calculate the area of each reef (delimited by contours) obtained from SIMAR ArrecifeSAM-CONABIO[66,67] (https://simar.conabio.gob.mx). Warmer colors indicate higher percentages of $CaCO_3$ loss after the mass mortality event caused by stony coral tissue loss disease (SCTLD; for more information, see Fig. 3). Stars indicate the sites where coral cores and fragments were collected. The sites Radio Pirata and La Catedral are part of the PM La Catedral reef unit. **b**–**e** Comparison of X-ray images and mean annual density values of all slabs obtained from living vs. dead coral colonies of each species analyzed in this study. This change was used to obtain a global indicator of $CaCO_3$ loss for each species. The X-ray images show that living colonies have higher density areas than those of dead colonies. The pie charts show the density loss (%) for living and dead colonies.

The LMM models created for each species also revealed a highly significant relationship between positions across the colony (progressive changes from recent density bands to older bands) and status (i.e., living or dead) for *D. cylindrus* ($x^2 = 19.40$, df = 1, $p < 0.01$), *P. strigosa* ($x^2 = 864.35$, df = 1, $p < 0.01$), and *S. siderea* ($x^2 = 6.99$, df = 1, $p < 0.01$). These findings indicate that the rate of density loss in dead colonies decreased from the colony surface toward the deeper areas of the skeleton (Fig. 2i–k). No such effect was observed for *O. faveolata* ($x^2 = 0.19$, df = 1, $p = 0.66$; Fig. 2l).

**Reef-scale losses of $CaCO_3$ after the 2018 SCTLD outbreak.** According to our estimations, the widespread mortality of coral colonies led to an enormous reduction in fixed $CaCO_3$ at the reef scale (Fig. 3). The standardized erosion rates are equivalent to area-wide total net losses of $CaCO_3$ ranging between −140 and −1471 t $CaCO_3$ km$^{-2}$, depending on the reef site, which can be exclusively attributed to the dissolution of the dead skeletons of *D. cylindrus*, *P. strigosa*, and *S. siderea* (Fig. 3a–c). The total amount of $CaCO_3$ lost across the shallow habitat of the entire reef system (~3,145,967.15 m$^2$) was equivalent to −4184 t, which represents an average dissolution of −1.33 kg $CaCO_3$ m$^{-2}$. The $CaCO_3$ losses attributable to these three species represent a

reduction of 6.78% of the total $CaCO_3$ fixed by all scleractinian coral colonies in the entire reef system after one single mortality event (Fig. 3d).

The loss of carbonate mass at each reef site depended on the abundance and size structure of the colonies of the various coral species present (Fig. 3a, b). For example, in Bonanza, 80% of the sequestered $CaCO_3$ was contained in colonies of *P. strigosa* and *S. siderea* (Fig. 3b). In contrast, the colonies of other scleractinian species were rare and small in size. Therefore, the relative losses in Bonanza following the SCTLD die-off were considerable (Fig. 3d). In contrast, in the reef units of Limones, La Catedral, and Tanchacte, the contributions of *P. strigosa*, *S. siderea*, and *D. cylindrus* to the amount of fixed $CaCO_3$ were relatively minor, as other coral species (e.g., *O. faveolata* and those that are not highly susceptible to SCTLD like *P. astreoides*, *A. palmata*, and *A. tenuifolia*) dominated these sites, resulting in comparatively minor net losses due to skeletal dissolution in these reef units (Fig. 3a–c).

**Discussion**

Our results reveal significant losses in the mass of dead coral colonies following skeletal exposure and provide quantitative insights into how living coral tissues prevent losses of the reef

**Table 1 Reef site, zone, date, and health status of coral fragments and coral cores at the time of collection.**

**A**

| Reef | D. cylindrus | | | | | | |
| | Catedral | | | Pared | | R. Pirata | |
|---|---|---|---|---|---|---|---|
| State | L | D | D | L | D | L | D |
| Year | 15 | 19 | 20 | 15 | 20 | 15 | 20 |
| Exposure | 0 | 1 | 2 | 0 | 2 | 0 | 2 |
| Replicates | 4 | 3 | 1 | 1 | 1 | 1 | 1 |

**B**

| | P. strigosa | | | | S. siderea | | | | O. faveolata | | |
| | Back | | Fore | | Back | | Fore | | Back | | Fore |
|---|---|---|---|---|---|---|---|---|---|---|---|
| Zone | | | | | | | | | | | |
| State | L | D | L | D | L | D | L | D | L | D | n/a |
| Year of sample | 20 | 20 | 20 | 20 | 20 | 20 | 20 | 20 | 20 | 20 | n/a |
| Exposure | 0 | 2 | 0 | 2 | 0 | 2 | 0 | 2 | 0 | 2 | n/a |
| Replicates | 4 | 3 | 3 | 2 | 6 | 3 | 4 | 4 | 4 | 6 | n/a |

Number of fragments collected from each colony of *Dendrogyra cylindrus, Pseudodiploria strigosa, Siderastrea siderea,* and *Orbicella faveolata*. Only *S. siderea, P. strigosa,* and *O. faveolata* samples were collected. 2015 = Living *D. cylindrus* colonies. 2019 = 1-year dead *D. cylindrus* colonies. 2020 = 2-year dead *D. cylindrus* colonies. The *D. cylindrus* colony at La Pared is very large [diameter (5 m); height (2 m)], which allowed for multiple samples to be collected. *L* Living; *D* Dead; 15 = 2015; 19 = 2019; 20 = 2020. Exposure indicates the number of years after the colonies died. *n/a* not applicable.

matrix. This finding is particularly relevant in the context of mass mortality events, such as those associated with disease outbreaks or temperature-induced bleaching events, as skeletal dissolution will weaken the calcareous structures of dead corals, threatening the long-term stability of the reef matrix. The $CaCO_3$ losses attributed to the widespread mortality of the colonies of only three coral species represented a reduction of almost 7% of the total amount of $CaCO_3$ fixed in the carbonate skeletons of the living corals of all species in our study system (Fig. 3d). This loss, which was due to skeletal dissolution and occurred in less than 2 years, is substantial, considering that the accumulation of $CaCO_3$ in these skeletons occurred over tens or even hundreds of years.

Our study highlight the impact of widespread coral die-off in terms of $CaCO_3$ loss over large geographical areas and bring the potential implications of SCTLD outbreaks at the regional level into perspective. However, it is crucial to consider that our findings only reflect the first of many destructive processes that follow coral mortality[16]. Therefore, we can expect that total $CaCO_3$ loss over the long-term will be notably higher than what we have quantified. This is particularly important given that mass loss makes coral skeletons more susceptible to other erosive and destructive forces. For example, weakened coral skeletons are more vulnerable to fragmentation following tropical storms[39] and are likely to be rapidly eroded by macroborers[33]. In the long term, the net result of the skeletal dissolution rates observed in this study will be a reduction in the structural complexity of the reef framework, which will likely affect the ecosystem services of the coral reefs, such as providing protection and food for commercially important fish species (Supplementary Fig. 2;[3,4]).

With our study, we provide quantitative insights into how live coral cover prevents loss of reef matrix, and how quickly coral skeletons erode once they become exposed. Mortality of living coral tissues caused by SCTLD causes changes in the composition of the microendolithic community from one that can interact positively with the coral to a purely eroding community (Supplementary Fig. 3; [21,40]). Thus, the loss of $CaCO_3$ observed in the dead coral skeletons in this study was due to a heightened increase in net

dissolution following the loss of protective tissue cover, microperforation, and the metabolic activity of the epilithic and endolithic algal and bacterial communities that colonized the skeletons[19]. The lack of protection by living tissues, coupled with the naturally high surface area of scleractinian coral skeletons, created conditions that were favorable to $CaCO_3$ dissolution[20]. This process can be classified as a type of succession. Initially, exposed skeletons are almost exclusively dominated by microborers that modify the substrate, making it accessible to macroborers and grazing epilithic bioeroders, which become increasingly important over time (e.g. refs. [16,41–43]). In turn, the bioeroders increase the surface area of the substrate by creating internal cavities and removing alternative surface covers, which creates conditions that favor passive dissolution[41,43].

One noticeable outcome from our analysis is that rates of $CaCO_3$ loss vary among coral species, which was reflected in the net carbonate loss observed across reefs with different species and colony sizes (Fig. 3c, d). The most likely explanations for the variation in net dissolution among different species are the differences among growth forms and skeletal microstructure arrangements[20,44]. We observed a clear difference in skeletal density between living and dead corals in species with meandroid corallite (*D. cylindrus* and *P. strigosa*) and cerioid (*S. siderea*) arrangements, with meandroid corals losing more material than ceroid corals. No significant differences between dead and living *O. faveolata* colonies, which exhibit plocoid corallite arrangements, were observed (Fig. 2h). The main difference between corallite types is the degree of separation between corallites by a $CaCO_3$ wall (see ref. [44]). While the walls between meandroids are incomplete, corallites in ceroid arrangements share walls. Plocoid corallites are separated by thick walls that are spaced apart.

In addition, the coral colony growth form also seems to influence the rates of skeletal dissolution. The columnar species *D. cylindrus* underwent the highest mass loss when compared to that of *P. strigosa*, which has a massive morphology (Fig. 2a, b). Overall, corallite arrangement and morphology influence the surface and internal areas available for microborer colonization and light penetration into the interior of the skeleton[23,45,46]. Therefore, colonies with porous skeletons facilitate the settlement of microendoliths, and small volumes favor their abundance by creating spaces with optimum light conditions[20,47]. This explains why we did not find any change in the density of *O. faveolata* skeletons, as its plocoid corallite arrangement reduces access to microborers, and its massive morphology results in an unfavorable surface:volume ratio when compared to those of other species[48]. These results suggest that the erosion rate of the reef framework strongly depends on the reef species. Coral communities dominated by species with high surface:volume ratios (i.e., branching, foliose, columnar, and sub-massive corals) and porous skeletons (i.e., meandroid corallites) will tend to erode faster than reefs dominated by massive species with less porous skeletons (i.e., plocoid corals; Fig. 3c, d).

We also found evidence that coral skeleton erosion did not occur linearly across time. Following the mortality event of 2018, we resampled the same *D. cylindrus* colonies 1 and 2 years later and found that skeletal erosion was not strongly reduced in the second year (Fig. 2a). This is consistent with previous studies indicating that the dissolution rate driven by microbiological activity does not scale over time and instead reaches a plateau a year after substrate exposure[16,43,49]. This pattern can be explained based on the succession of endolithic communities, which is highly dynamic over time[16,23,43]. Contrary to our observations, Enoch et al.[50] reported gains in substrate density (instead of carbonate dissolution) in experimental substrates deployed on acidified reefs after 2 years. However, the authors did not rule out that dissolution processes might have preceded cementation[51].

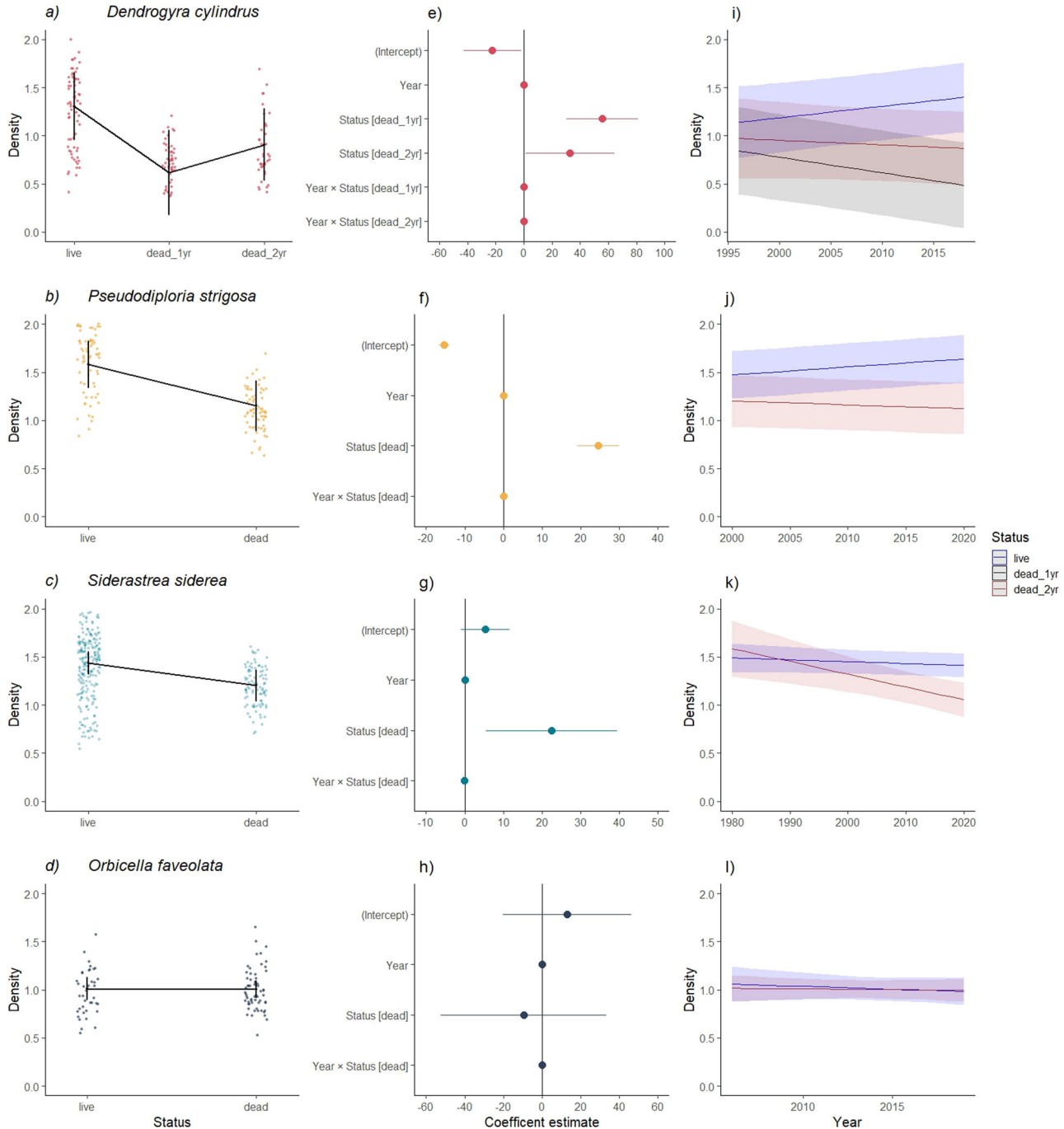

**Fig. 2 Changes in density and total mass between living vs. dead colonies.** The plots on the left (**a–d**) show the absolute mass change between living vs. dead colonies. The dots indicate the annual density values calculated for each species according to its state (i.e., living or dead). The black lines above the dots indicate the 95% CIs. The central graphs (**e–h**) show the coefficient estimates and confidence intervals of the linear mixed models (LMM). The dots represent the average slope of each model and lines represent coefficient and indicate the 95% CIs. Slopes are significantly different from zero if their 95% CIs do not overlap with the vertical dashed line centered on zero. All plots were derived from the LMM estimates of linear trends. The plots on the right (**i–l**) show trends for each species for changes in density in different sections of the colonies (current years refer to shallow sections of the colony; older years indicate deeper sections of the skeleton). Shading represents the 95% CIs. All plots are based on LMM models.

While the existing evidence suggests an absence of linearity in dissolution processes (see refs. [16,49] and Fig. 2a), it is still unclear how the succession of dissolution and cementation occurs, and studies are needed to evaluate these processes in different environments (e.g., oligotrophic and acidified sites) with different skeletal types and over different time scales.

Mass loss in dead corals was highest at the surface and lessened toward the deeper areas of the skeleton. This trend was most evident in *D. cylindrus*, although it also occurred in *P. strigosa* and *S. siderea* (Fig. 2i–k). Higher rates of erosion on the colony surface suggest that bioerosion was dominated by microborers, most of which are photosynthetic and require favorable light conditions (e.g. ref. [25]). For instance, the characteristic pillars of *D. cylindrus* exhibit less volume in their apical zones and more volume at their bases, which might explain why the highest difference in density was observed between the living

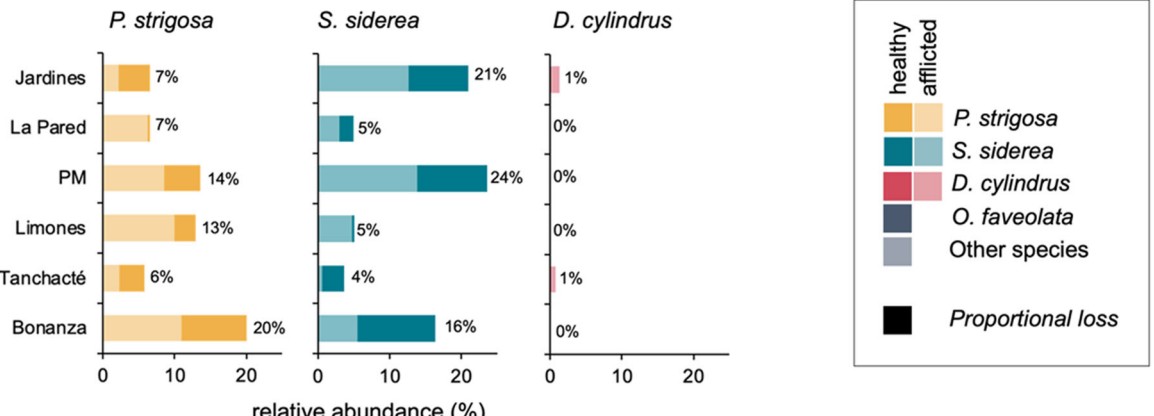

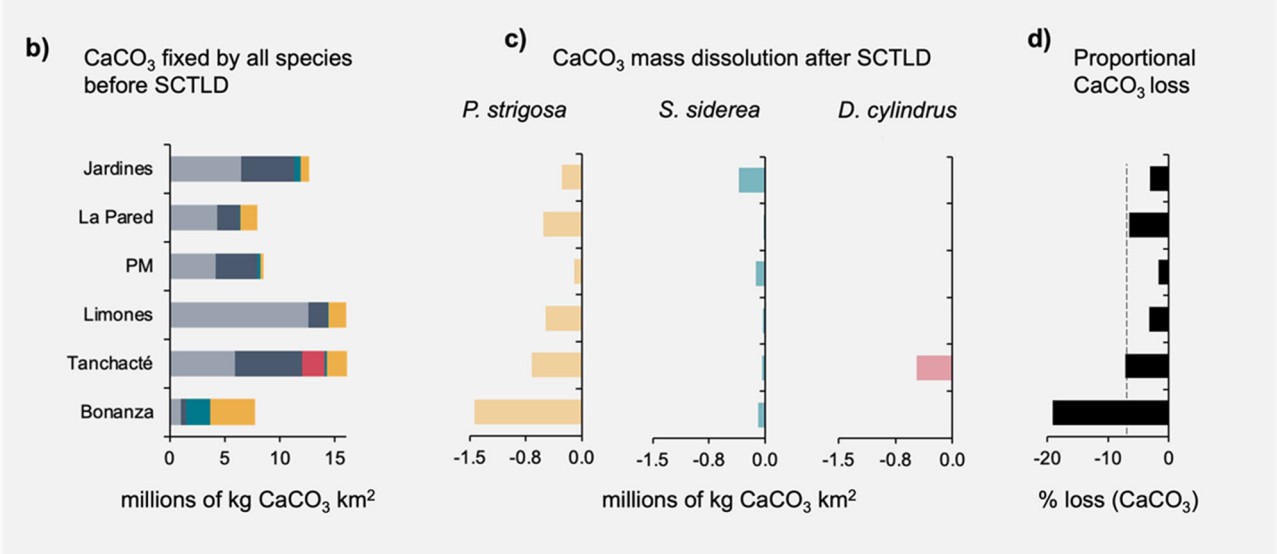

**Fig. 3 Loss of CaCO₃ after the mass mortality caused by stony coral tissue loss disease (SCTLD) in the Puerto Morelos reef system. a** Relative abundance and the proportion of colonies of *Pseudodiploria strigosa*, *Siderastrea siderea*, and *Dendrogyra cylindrus* afflicted by SCTLD in each reef in 2019. The numbers in front of the bars indicate the relative abundance (%) of each species compared to all species surveyed in each reef. **b** Total CaCO₃ fixed in the coral colonies of all species before the SCTLD outbreak. **c** Estimated mass loss by dissolution at the reef scale 2 years after the death of the colonies due to SCTLD in each reef. **d** Reef-scale proportional loss of CaCO₃ by dissolution in the skeletons of the three species studied in each reef unit (considering all CaCO₃ contributed by all the species). The dotted line indicates the mean percentage lost at the reef system level (7%). These quantities were obtained from the average estimates calculated for each species in the transects ($n = 87$) and multiplied by the area of each reef (m²).

and dead samples collected from the superficial zones of the colony. Our findings also indicate that microbioerosion in dead corals can occur down to ~7 cm (Fig. 2i–k). Identifying the specific agents that affected net dissolution in the different sections of the dead skeletons is beyond the scope of this study. However, evidence from previous studies suggests that microbioerosion depends on the penetration capacity of microboring organisms[16,52]. Although high microbioeroder biomass has been reported within a few millimeters of the surface (~4 mm; [49]), this result was based on observations of pristine blocks constructed from coral and bivalve skeletons (Supplementary Table 3). The penetration capacity of microborers is likely underestimated with blocks, as these experimental structures reflect only young communities growing on new substrates, which require time to bore deeper[23,43,49]. In addition, rapid surface colonization on bioeroder-free substrates interferes with the colonization of endolithic communities in deeper layers[53]. In addition, the pH of

coral porewater is lower than that of ambient water and produces physicochemical gradients that create corrosive conditions suitable for CaCO₃ dissolution[24,30,54]. Additional experimental and in situ studies are needed to understand which agents and processes dissolve CaCO₃ in the different depth layers of coral skeletons.

Some experimental studies on carbonate dissolution have been conducted to test the effects of acidified environments, with one such study reporting dissolution rates of $-1.52 \pm 0.08$ kg CaCO₃ m⁻² year⁻¹ [53] (see Supplementary Table 3). However, our study shows that net dissolution processes were high ($-1.33$ kg CaCO₃ m⁻²) in our study area, even in reefs that did not reflect signs of low pH or high temperatures (see ref. [54] and Supplementary Methods; Appendix 1, Supplementary Table 5). This suggests that the net dissolution of dead skeletons is mainly dominated by the influence of microborers and associated biogeochemical processes (e.g., the respiration and remineralization of

endolithic communities) rather than the low pH levels of the ambient water. Nevertheless, in a scenario combining mass mortality, low pH levels, and high temperatures, calcium carbonate losses could be even more severe with rates as high as $4.92 \pm 0.3$ to $19.00 \text{ kg CaCO}_3 \text{ m}^{-2} \text{ year}^{-1}$ (refs. [19,24]; for the purposes of comparison, these experimental dissolution rates were transformed; see Supplementary Table 3).

Our study used densitometry analysis to measure in situ mass loss in adult corals after colony death. This approach captures the effects of erosive processes in coral colonies exposed to prolonged bioeroder succession, thus mimicking natural conditions better than approaches employing coral blocks[16]. For example, the infestation of endolithic communities in pristine experimental blocks can only occur from the outside in. In contrast, infestations can also occur from the inside out in recently deceased coral colonies due to pre-existing communities blooming within the skeletons[19,20,23]. Furthermore, in contrast to other widely used methods that have been employed to measure $CaCO_3$ erosion rates, such as volumetric or buoyant weight analyses, optical densitometry allows detailed analysis of the properties of $CaCO_3$ across coral cores for time series, as well as a focus on microborers because macrobioerosion can be avoided[50,53].

Overall, our findings highlight the notable impact of mass coral mortality on framework loss at the ecosystem level and the vulnerability of Caribbean reefs following an SCTLD outbreak. These losses weaken coral skeletons, making colonies more vulnerable to additional threats that jeopardize the integrity and structural complexity of the reef framework (Supplementary Fig. 2;[3,4]). However, the patterns and degree of carbonate mass loss depend on the species composition of each reef. Small-volume corals with porous corallite arrangements are likely to erode more than reefs dominated by massive species with less porous corallite arrangements (Fig. 3;[46]). This finding is particularly critical given that most Caribbean reefs have shifted from being dominated by large, massive reef-building species (e.g., *Orbicella* spp.) to being dominated by small species that contribute relatively low amounts of $CaCO_3$[38,55]. Furthermore, our findings highlight the importance of including the biological and chemical processes of $CaCO_3$ dissolution in carbonate budgets[18,19], especially as the impacts from global warming, ocean acidification, and disease enhance dissolution processes and increase substrate availability for colonization by microeroders[37].

## Materials and methods
### Site and sample collection
The Puerto Morelos reef system is located near the northeastern portion of the Yucatan Peninsula in Mexico and is comprised of separate shallow reef units. The reef system covers 9066 ha and extends over 20 km. The main coral species represented in the shallow reefs are *A. palmata*, *O. faveolata*, *O. annularis*, *P. strigosa*, *S. siderea*, *Agaricia spp.*, and *Porites spp.*[56,57]. Each reef unit comprises a sheltered back reef zone and an exposed fore reef zone separated by a crest. In June 2018, the Puerto Morelos reef system was afflicted by an SCTLD outbreak that resulted in widespread coral mortality[36,58].

To measure mass loss in the skeletons after coral death, we collected colony fragments and coral cores from four species: *D. cylindrus*, *P. strigosa*, *S. siderea*, and *O. faveolata*. These species were selected because (1) they are conspicuous species in Puerto Morelos; (2) they are highly susceptible species to SCTLD[58]; and (3) we were able to identify (within ~15 days) the date of mortality of the sampled colonies of each species, as we had other ongoing studies in the area (see refs. [33,36,38,59]).

We used two different sampling designs, as previously collected samples of *D. cylindrus* were available. In 2015, *D. cylindrus* colonies that were living with no signs of disease were identified and sampled[59]. In this study, we repurposed and reprocessed these previous samples and sampled the same colonies two additional times: 1 year after the total mortality event (2019) and 2 years after the mortality event (2020; Table 1 and Fig. 1a). For the other three species (*P. strigosa*, *S. siderea*, and *O. faveolata*), we sampled living and dead colonies in 2020 (Table 1 and Fig. 1a). All cores from dead colonies were obtained from colonies that were known to have died due to SCTLD, and the date of mortality was recorded (within 15 days). In this case, all samples from dead colonies reflected 2 years of exposure. The samples were obtained from different colonies found within an area of ~30 m² in La Catedral (Fig. 1a). Samples from all colonies (living and dead) were collected from the apical part of the colony. Fragments obtained from *D. cylindrus* colonies were ~10 cm in diameter and 10–20 cm in length (Table 1A). Cores obtained for *P. strigosa*, *S. siderea*, and *O. faveolata* were 3 cm in diameter and 5–15 cm in length. Data for all colony fragments and coral cores are shown in Table 1B.

To contextualize the possible mass loss of *D. cylindrus*, *S. siderea*, *P. strigosa*, and *O. faveolata* with the $CaCO_3$ contributed by the coral community, we obtained fragments and cores of the most representative scleractinian species within the Puerto Morelos reef system. All samples were collected at La Catedral (*Acropora palmata* = 2, *Agaricia agaricites* = 4, *Agaricia teunifolia* = 4, *Orbicella annularis* = 4, *Porites astreoides* = 8, *Pseudodiploria clivosa* = 3, *Porites porites* = 4, *Stephanocoenia intersepta* = 2, *Montastraea cavernosa* = 4, *Dichocoenia stokesii* = 6, *Colpophyllia natans* = 3, and *Siderastrea radians* = 4). For all samples of these species, skeletal density was calculated as described below.

### Changes in skeletal density and porosity between living and dead coral colonies
Each fragment and coral core was cut into 9-mm thick slabs following the main growth axis using a rock saw equipped with a diamond blade (Fig. 1b–e). Subsequently, all slabs were dried in an oven at 60 °C for 48 h. In March 2020, all slabs were radiographed using a CR Medical Systems X-ray machine (GE-Healthcare, Chicago, IL, USA). The slabs were oriented longitudinally in rows for radiographic scanning (2–3 slabs per row), alternating dead and living slabs. The parameters for each scan were 55 Kv and 6 mAs for all exposures, and the pixel size was 100 μm². During each scan, the slabs were accompanied by an aragonite standard that consisted of a wedge of known thickness and density (2.83 g $CaCO_3$ cm$^{-3}$) according to the methodology in Carricart-Ganivet and Barnes[60].

Subsequently, the alternating density bands were identified in digitized X-radiographs. The aragonite standard was used to create a calibration curve from which the annual density of each slab (g $CaCO_3$ cm$^{-3}$) was determined. Annual density series (g cm$^{-3}$) were obtained from the linear distance along the slabs from the apical zone (recent growth years) to the base of the slabs (old growth years). We used the optical densitometry method described in Carricart-Ganivet and Barnes[60]. The extension rate (cm year$^{-1}$) was determined from the distances between density minima peaks. Following this method, we obtained annual density and extension values (one for each pair of bands). The annual density values obtained from the growth bands was considered the units of analysis for this study (see more details in the Statistics and reproducibility section).

To obtain the porosity data, each digital X-ray image was segmented using the intensity of the gray color of each slab following the methodology of Fordyce et al.[31] with modifications. For this, the minimum and maximum thresholds of gray intensity

(0–255) were defined using the *Threshold function* in ImageJ v. 1.8.0 (Supplementary Fig. 4). The pixels within each slab that had the same values as the average gray value of the aragonite standard were assigned as the maximum threshold ($S_{max}$). The pixels within each slab with the same average value as that of the image background (i.e., air) and the first 0.5 mm of the wedge (intermediate air/CaCO$_3$ phase) were established as the minimum threshold ($S_{min}$). All of the pixels within each slab that had values between $S_{max}$ and $S_{min}$ corresponded to porous spaces. Once the thresholds were adjusted, the ImageJ function *Analyse Particles* was used on the segmented image to obtain the number of porosities found throughout the slab (Supplementary Fig. 4). The gray values of the entire slab generated during thresholding were averaged and subsequently calibrated using the curve made from the aragonite wedge to obtain the average total density of each slab, which was used to complement the corresponding porosity percentage information.

Differences between the physical properties of living vs. dead coral skeletons were assessed through optical densitometry and porosity analyses. Optical densitometry relates X-ray attenuation to skeletal density along the main colony growth axis[60]. The porosity analysis offered information about air spaces according to the spatial resolution of all parts of the slabs[31]. Together, changes in skeletal density and porosity after the death of a colony reflect the net outcome of biotic and chemical dissolution processes on a finer scale than those detected using only porosity analyses or other volumetric quantification methods[50]. In this study, we were only interested in measuring density changes in coral skeletons resulting from net erosion (microerosion and dissolution processes); therefore, we only measured sections of the coral cores that had not been eroded by internal macroborers (e.g., bivalves, and worms).

**Statistics and reproducibility.** The annual growth data obtained from coral cores (6–38 per core, see Supplementary Table 1), has a relatively complex hierarchical structure because the individual sampling units are coral cores (12–17 cores per specie, see Table 1). The observational units used for the statistical analyses are the density estimates obtained from the multiple annual coral growth bands that were estimated from the individual cores. The sampling of annual growth bands is not random, as annual coral growth bands derived from the same cores are likely correlated.

To analyze our data, we followed an approach used in previous studies dealing with a similar data structure (e.g. refs. [61,62]). We used a random intercept Linear Mixed Model (LMM) with residual correlation structures to model skeletal density as a function of colony status (i.e., living or dead). In addition, we also considered the year of each annual growth band as a fixed factor to test for differences in the skeletal density across the coral cores (i.e., from the surface to deeper areas of the skeleton). Colony identity, reef area (back-reef or fore-reef), and the annual extension rate (i.e., extension between growth bands across each core) were treated as random effects. This approach distinguishes observational units from sampling units, recognizes that sampling variation exists within and between core time series, and addresses the temporal autocorrelation structure inherent in such data[62]. All models were constructed using the *lme4* package in R[63].

We used a separate LMM for each species because different sampling designs were used for *D. cylindrus* and the other three species (see above and Supplementary Methods; Appendix 2, Tables 6–13). The differences in each species' skeletal structure and growth strategy likely increase model uncertainties when lumping all the data into one model. However, we also ran two

exploratory models combining the data of all four species (see Supplementary Methods; Appendix 2, Tables 14–16 and Supplementary Figs. 5 and 6), and only the three species (see Supplementary Methods; Appendix 2, Tables 17–19 and Supplementary Figs. 7 and 8), for which the data were obtained in the same manner to confirm that a similar pattern was derived when integrating data from different species. As expected, the level of uncertainty also increased.

The goodness of fit of each LMM was evaluated with the *DHARMa* package via an analysis of residuals. To test for differences in skeletal density between living and dead colonies, an analysis of variance (Type II Wald chi-square tests) was performed for each model with the function *Anova* of the *car* package. Since the *D. cylindrus* model had three sampling levels (i.e., living colony, 1 year after colony death, and 2 years after colony death), a test of individual contrasts using the *lsmeans* package was used to evaluate differences between the three levels. All analyses were performed in R[63].

Raw data are accessible in Supplementary Information. The methods for statistical analysis and sizes of the samples (defined as *n*) are given in the respective sections of results and methods. Statistical analyses were conducted in R Version 3.6.173 using the cited packages, and reproducibility can be achieved using the parameters reported in this section.

**Benthic surveys and CaCO$_3$ loss at the reef scale.** We estimated the loss of CaCO$_3$ associated with the mortality of *D. cylindrus*, *S. siderea*, and *P. strigosa* colonies due to the SCTLD outbreak in the Puerto Morelos reef system and contrasted this carbonate loss with the CaCO$_3$ contributed by all colonies of other coral species that were not affected by the outbreak. For this, we used the estimates of skeletal density obtained from living and dead colonies and field surveys.

Field data were collected from July 2018 to September 2019 (during the SCTLD outbreak) from six reef sites across the reef system (Fig. 1a). Coral communities were surveyed at each reef site using 10 to 25 randomly placed belt transects (10 × 1 m). The following data were recorded for each coral colony within each transect: species identity, colony size (maximum diameter, diameter perpendicular to the maximum diameter, and height), bleaching percentage, mortality percentage (recent, transient, and old), and the presence of SCTLD or other diseases (see ref. [36]). We also recorded colonies with 100% mortality that could be attributed to SCTLD (i.e., recent mortality was still evident[38]).

In the lab, we calculated the volume of all surveyed colonies following different approaches according to their growth. For the massive and sub-massive species (*D. cylindrus*, *P. strigosa*, *S. siderea*, *O. faveolata*, *O. annularis*, *P. astroides*, *P. clivosa*, *S. intercepta*, *M. carvernosa*, *D. stokessi*, *C. natans*, and *S. radians*), we used the length, width, and height, according to the equation for elliptical paraboloid growth forms proposed by González-Barrios and Alvarez-Filip[64]:

$$C = \left(\frac{\pi}{2} H r^2\right), \tag{1}$$

where *C* is the volume of each colony, *H* is the height of the colony, and $r^2$ is the mean colony diameter.

To calculate the volume of the colonies of the branched and foliose species (*A. palmata*, *A. cervicornis*, *P. porites*, and *Agaricia* spp.), we used the following workflow: (1) we obtained the volume from 3D models built by photogrammetry of *A. palmata* colonies in situ ($N = 7$) and through 3D models made with an EinScan SE model scanner (Shining 3D, Hangzhou, China; pressure 0.05 mm) of *A. palmata* skeletons from different collections ($N = 24$). (2) We measured the height, minimum

diameter, and maximum diameter of the colonies used for the 3D models ($N = 31$). (3) These metrics were substituted into the volume equation of a rectangular prism. (4) We compared the volumes obtained from the 3D models and the volumes calculated by the rectangular prism equation to calculate a correction factor (0.11). (5) We calculated their volume by applying the correction factor to the rectangular prism equation from the data obtained during our surveys of the branching colonies (height, minimum and maximum diameters).

We then used the volume to calculate the amount of $CaCO_3$ fixed in each colony by multiplying species density by the volume of each colony, corresponding to the carbonate mass attributed to living colonies of *D. cylindrus*, *S. siderea*, and *P. strigosa* with no visible SCTLD lesions (or those covering <10% of the colony surface); all *O. faveolata* colonies (as no loss of density was observed between dead and living colonies; see results section); and those of all other species.

We calculated two scenarios for dead or afflicted colonies of *D. cylindrus*, *S. siderea*, and *P. strigosa*. First, assuming the colony was alive, it was assigned the density obtained from the living colonies (Eq. (2); Fig. 3c). Second, assuming each colony was dead, it was assigned the density obtained from the dead skeletons (Eq. (3); Fig. 3d). We assumed that all diseased colonies with SCTLD lesions covering >10% of the colony surface would have died[65]:

$$M_l = C.D_l \qquad (2)$$

and

$$M_d = C.D_d \qquad (3)$$

where $M_l$ is the $CaCO_3$ mass of each living colony, $M_d$ is the $CaCO_3$ mass of each dead colony, $H$ is the height of the colony, $r^2$ is mean colony diameter, $D_l$ is the average density of each coral species calculated from living skeletons (Fig. 1b–e and Supplementary Table 3), and $D_d$ is the average density of each coral species calculated from dead skeletons (Fig. 1b–e and Supplementary Table 2).

For dead or afflicted *D. cylindrus*, *P. strigosa*, and *S. siderea* colonies, we estimated kg $CaCO_3$ m$^{-2}$ for a "pre-SCTLD" scenario, in which all coral colonies were alive, and a "post-SCTLD" scenario, in which all colonies with SCTLD died and thus underwent $CaCO_3$ losses. For both scenarios, the $CaCO_3$ contributed by each colony was added and then divided by the survey area of each reef (Fig. 3c, d). The difference in kg $CaCO_3$ m$^{-2}$ between the "pre-SCTLD" and "post-SCTLD" scenarios was interpreted as the mass loss caused by biological and chemical dissolution following SCTLD-induced mortality.

Lastly, we used the above estimates of $CaCO_3$ loss to predict the potential broad-scale consequences of biological and chemical dissolution for the entire reef system of Puerto Morelos. We obtained the area of all reef units from SIMAR ArrecifeSAM-CONABIO (see Supplementary Table 4;[66,67]), a database that integrates data from WorldView-2 satellite images of the spatial distributions of benthic habitats in the shallow-water marine ecosystems of the Mexican Caribbean (2010–2018) that have been validated with field data (Fig. 1a). To determine the loss of $CaCO_3$ in the entire Puerto Morelos reef system, the amount of $CaCO_3$ contained in each reef was calculated for each period (before and after the SCTLD outbreak). The difference between periods was considered to be the total $CaCO_3$ loss at the reef scale attributable to the mortality of *D. cylindrus*, *P. strigosa*, and *S. siderea* colonies following SCTLD-induced mortality.

**Reporting summary.** Further information on research design is available in the Nature Portfolio Reporting Summary linked to this article.

## Data availability
The authors declare that the raw data and numerical source data for the graphs are available in the GitHub repository: https://github.com/FRAME57/Medellin-Maldonado_2023_CommsBiol_2.

## Code availability
The authors declare that the R code are available in the GitHub repository: https://github.com/FRAME57/Medellin-Maldonado_2023_CommsBiol_2.

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

## Acknowledgements

F.M.-M. thanks the Postgraduate Department of Ocean Sciences and Limnology of UNAM for the opportunity to study and its support. We thank M. A. Gomez-Reali and E. Escalante-Mancera for assistance with the instrumentation of ambient loggers, and S.D. Guendulain-García for 3D scans and photogrammetry data. The comments and suggestions of two anonymous referees greatly improved our analysis and the manuscript. We also thank Andrea Lievana for editing the manuscript. This study was supported by the Mexican Council of Science and Technology (CONACyT, grant number: FORDECYT-PRONACES/425888/2020 to L.A.-F.), the Comisión Nacional de Áreas Naturales Protegidas of Mexico (grant number: PROREST/CER/56/2019 to L.A.-F.), and the Programa de Apoyo a Proyectos de Investigación e Innovación Tecnológica, UNAM (PAPIIT-IN209014 to J.P.C.-G.). F.M.-M. was supported by an PhD scholarship (no. 755995) from CONACyT.

## Author contributions

Conceptualization: F.M.-M. and L.A.-F.; Funding acquisition: L.A.-F. and J.P.C.-G.; Project administration: L.A.-F. and F.M.-M.; Writing original draft: F.M.-M. and L.A.-F.; Methodology and investigation: F.M.-M., L.A.-F., A.L.-P., E.P.-C., J.P.C.-G., I.C.-O. and O.N.-L.; Data curation: F.M.-M., L.A.-F., E.P.-C. and I.C.-O.; Formal analysis: F.M.-M., L.A.-F., I.C.-O. and O.N.-L.; Visualization: F.M.-M., L.A.-F. and A.L.-P.; Writing – review and editing: F.M.-M., L.A.-F., A.L.-P., E.P.-C., J.P.C.-G., I.C.-O. and O.N-L.

## Competing interests

The authors declare no competing interests.
