## [Peer Review File · Communications Biology]

Reviewers' comments:

Reviewer #1 (Remarks to the Author):

The manuscript entitled 'Coral skeleton dissolution: the other side of mass coral mortality' is a well written account of the potential impact of rates of coral dissolution on carbonate budgets (and reef state) following an outbreak of SCTL. I thoroughly enjoyed reading the paper as it was easy to read, provided a detailed overview of methods, well considered graphs (as well as supp material) and an interesting discussion which highlighted well the studies main findings and provided a clear significance of the results. I do, however, have 3 main issues related to: 1) the sample size and scaling up to the reef scale, 2) the representativeness of the corals that were studied and used to scale up the results, and 3) sample design which largely used different coral colonies for the live versus dead assessment. I provide more detail below on these three points as well as providing additional comments (and minor suggestions) in the attached PDF. Overall, I think that these three issues can be addressed as part of the review process. Given that this paper provides much needed data on a very understudied element of carbonate budgets and is therefore of significant value, I recommend the paper for publication following minor changes.

Sample size

A total of 51 coral colonies were analysed. This does represent a lot of work, but when split between the 4 coral colonies and time points – the sample size is quite small. I don't see this as a problem, but its important to take this into account when scaling up the results to large reef areas based on this number of coral colonies. Further, there did seem to be quite a wide variation in the densities between colonies, which typically means you need more coral colonies. Obviously, your stats suggests that there are significant difference, but I still think this needs to be at least commented on when then using these data to infer larger impacts.

Representativeness

From the paper, I have no idea what proportion of the coral cover is composed of these 4 corals. You make several statements throughout the paper that states the 'huge' impact of the associated carbonate loss - but you don't know that unless you understand the contribution of carbonates from the rest of the coral community. For example, even if your 4 corals have suffered a decline of 30% in carbonate production, if they only represent, 5% of the coral cover – the relative impact is quite low. So I would like to see the addition of coral cover data to get a feel of how important this carbonate loss is.

Further, only 10 m² were assessed in the surveys per reef, that were then used to scale up to much larger areas. So again how representative is this 10 m² in terms of coral cover to the rest of the reef? I worry that perhaps in looking for corals to sample – there might have been some bias as to where transects were placed because in my experience just writing 'a haphazard approach' was used – tends to mean that there were other (possibly) important reasons that used to choose a location for a transect such as logistics etc. So again, I would like to be sure that if you are using this 10 m² to then scale up to the reef – that you're accompanying statements are supported by data that shows that these sections are representative of the greater reef area.

Sample design

Almost all assessment of density for dead corals were conducted in different colonies from those that remained healthy. And looking at Figure 2 – the data (may) suggest that the colonies that remained healthy and not influenced by SCTL had higher densities than those that were infected. So this may then suggest that lower density colonies were more susceptible to the disease – which then saw a further lowering of the density. If this is the case, this would be an interesting finding, and would also mean that total loss of carbonate following the disease outbreak may have been over-estimated. i.e. if you applied average loss rates to average pre-densities as opposed to using colony specific data.

Reviewer #2 (Remarks to the Author):

As attached

Reviewer #1 (Remarks to the Author):

The manuscript entitled 'Coral skeleton dissolution: the other side of mass coral mortality' is a well written account of the potential impact of rates of coral dissolution on carbonate budgets (and reef state) following an outbreak of SCTLD. I thoroughly enjoyed reading the paper as it was easy to read, provided a detailed overview of methods, well considered graphs (as well as supp material) and an interesting discussion which highlighted well the studies main findings and provided a clear significance of the results. I do, however, have 3 main issues related to: 1) the sample size and scaling up to the reef scale, 2) the representativeness of the corals that were studied and used to scale up the results, and 3) sample design which largely used different coral colonies for the live versus dead assessment. I provide more detail below on these three points as well as providing additional comments (and minor suggestions) in the attached PDF. Overall, I think that these three issues can be addressed as part of the review process. Given that this paper provides much needed data on a very understudied element of carbonate budgets and is therefore of significant value, I recommend the paper for publication following minor changes.

R: We thank your positive assessment of our manuscript. We have carefully revised your comments and provided a detailed description of how we have amended the manuscript below each comment.

Sample size

A total of 51 coral colonies were analysed. This does represent a lot of work, but when split between the 4 coral colonies and time points – the sample size is quite small. I don't see this as a problem, but its important to take this into account when scaling up the results to large reef areas based on this number of coral colonies. Further, there did seem to be quite a wide variation in the densities between colonies, which typically means you need more coral colonies. Obviously, your stats suggests that there are significant difference, but I still think this needs to be at least commented on when then using these data to infer larger impacts.

R: We understand your concern. However, we would like to point out that while colonies (i.e., cores or fragments) were the sampling units, the observational units used to construct the statistical models, are the annual density data obtained from the multiple annual bands of coral growth. For this reason, there is a great variation in skeletal density in each species (reflecting that size and density of growth bands is influenced by annual environmental variables).

These annual density estimates were obtained from optical densitometry of each of the cores. On average, 14 annual estimates were obtained for each core (see supplementary information). The total sample size include in our models is therefore 654 (see dots in Figure 2 A and D). We used this approach as it gives a clear perspective of the density change in different sections within the skeletons as it is based on the analysis of bands along the cores (from the edge of the colony to the center of the colony). We would like to also note that this hierarchical structure of the data was considered in the construction of the linear mixed models. **We now included an appendix 2, which shows the number of observational units per core (Table S19). In addition, in this new version of the MS, we provide a detailed explanation of all this information in the methods section (Lines 471-509).**

Please also note that the number of replicates used in our study is well within the number of replications used in similar studies. We now provide a supplementary table (Table S2) summarising previous studies that have measured the substrates' net dissolution. This table is also helpful to represent that most previous studies have employed substrates constructed from the skeletons of mainly one coral genus (*Porites*) and are not necessarily

representative of how dissolution rates operate across different coral species. Our study, therefore, represents one of the first attempts that estimates the net dissolution rate for specific substrate types (i.e., coral species).

Representativeness

From the paper, I have no idea what proportion of the coral cover is composed of these 4 corals. You make several statements throughout the paper that states the 'huge' impact of the associated carbonate loss - but you don't know that unless you understand the contribution of carbonates from the rest of the coral community. For example, even if your 4 corals have suffered a decline of 30% in carbonate production, if they only represent, 5% of the coral cover – the relative impact is quite low. So I would like to see the addition of coral cover data to get a feel of how important this carbonate loss is.

R: Thank you very much for this excellent observation. Following your comment, we have included in our analysis the CaCO₃ contribution of all species surveyed in each reef. This allowed us to put in context the magnitude of the dissolution that occurred after SCTLD-induced mass mortality in the four species we studied. We have rewritten the results section and added the CaCO₃ contribution by all species in each reef unit (Fig. 3), to show the significance of mass loss in the studied coral species (*P. strigosa*, *S. siderea* and *D. cylindrus*).

The percentage of CaCO₃ lost attributable to the dissolution of our study species from the total CaCO₃ accumulated in the reef (i.e., considering all species as suggested) ranges from 3.07% and 19.12% (Fig. 3). Which continue to represent a major impact on the stability of the reefs in our study area.

Now in the methods section, we add the information in "*Benthic surveys and CaCO₃ loss at the reef scale*". Where we provide a detailed explanation of how the CaCO₃ of all species was incorporated into our analyses (lines 511-597):

Further, only 10 m² were assessed in the surveys per reef, that were then used to scale up to much larger areas. So again how representative is this 10 m² in terms of coral cover to the rest of the reef? I worry that perhaps in looking for corals to sample – there might have been some bias as to where transects were placed because in my experience just writing 'a haphazard approach' was used – tends to mean that there were other (possibly) important reasons that used to choose a location for a transect such as logistics etc. So again, I would like to be sure that if you are using this 10 m² to then scale up to the reef – that you're accompanying statements are supported by data that shows that these sections are representative of the greater reef area.

R: Thank you for pointing this out, as it gives us the opportunity to clarify this in the manuscript. We used between 10 and 25 transects of 10 m² per reef unit. This represents an area of 100 to 250 m² surveyed per site. In total we conducted 87 transects representing a total surveyed area of 870 m². The number of transects used to represent each reef is considerably higher to the used by other ecological studies or monitoring protocols which generally sample fewer transects (3 to 6 transects; see for example the AGRRRA protocol).

We now added the number of transects used to survey each reef unit. Lines 520-522 that "...Coral communities were surveyed at each reef site using 10-25 randomly placed belt transects (10 × 1 m)"

Sample design

Almost all assessment of density for dead corals were conducted in different colonies from those that remained healthy.

R: We agree with this observation. We recognize that using different colonies exerts an effect on the variability of skeletal density. However, we account for this effect by considering the colony identity as random factor in the Linear Mixed Models.

Please note that the effect of colony identity is deemed to have a minimal contribution to the explained variable of the models (see appendix 2 model selection diagnostics), which shows that that colony identity does not have a significant effect on the difference in skeletal density between live and dead colonies. Moreover, please also note that the time of exposure to bioerosion processes for all species in the dead colonies was approximately the same (two years)

Following this comment and the “Statistics comment” of the second reviewer, we now provide in the methods section a detailed justification for the use of separate models for each species in lines 471-509. We have also added to the supporting material (appendix 2) a rationale for the construction of the model for each species and the summary results

And looking at Figure 2 – the data (may) suggest that the colonies that remained healthy and not influenced by SCTLD had higher densities than those that were infected. So this may then suggest that lower density colonies were more susceptible to the disease – which then saw a further lowering of the density. If this is the case, this would be an interesting finding, and would also mean that total loss of carbonate following the disease outbreak may have been over-estimated. i.e. if you applied average loss rates to average pre-densities as opposed to using colony specific data.

R: Many thanks for this observation. The hypothesis that colonies with lower skeletal density were more susceptible to disease due to weakening prior to the SCTLD outbreak seems to be plausible. However, for the species we have information of the same colony before and after mortality (*D. cylindrus*) we also found a strong signal of decreasing skeletal density after mortality, which support our interpretation that skeletal density was reduced by dissolution following mortality. Furthermore, if skeletal density would be associated with the propensity to SCTLD infection, this would be also reflected in *O. faveolata*, as this species is also highly susceptible to the disease. And, for this species we found no significant differences in skeletal density between dead and live colonies.

Minor comments

1. Abstract well written. Highlights key findings and significance. My only suggestion would be to provide some quantitative data to support key statements.

R: Thank you, we have added in lines 21-24

“...We observed that the average dissolution rate of the entire reef system was $-1.33 \text{ kg CaCO}_3 \text{ m}^{-2}$, which means a reduction of CaCO_3 sequestered in the skeletons of *P. strigose*, *S.siderea* and *D. cylindrus* from 23.90% to 33.91% in each reef unit.”

2. Line 45 remove future

R: Done (line 49)

3. Line 54-60. Well there are over 30 studies that have put together detailed carbonate budgets which include destructive processes albeit - the confidence around some of the assessment of rates of carbonate loss is less because it is so difficult to measure accurately

R: Agree; many studies include erosive processes in their carbonate balances. Nevertheless, some of these carbonate balance use values that come from other studies. That is because erosional processes are much less well understood, mainly because of the difficulty of accurately measuring these processes. For this reason, these lines (58-64), intend to highlight a knowledge gap that still exists.

4. Line 61- yes but - there are several studies that have measured bioerosion rates e.g. Pat hutchinson and her team which I think is worth a mention here.

R: We agree that Hutchings' work should be referenced because of his significant contributions to the field. We do cite the studies that have measured external bioerosion rates in line 58-63. However, in this line (now line 64), we refer only to microbioerosion and chemical dissolution to emphasise that these are the least understood processes. We have reworded the text to make this point clear (lines 64-70):

"...However, the consequences of dissolution, especially those that follow coral mortality, are poorly understood (Grange et al. 2015; Schönberg et al. 2017), as most are complex and involve multiple taxa and environmental factors (Glynn 2015; Schönberg et al. 2017, Browne et al. 2021). Skeletal dissolution can be driven by microboring organisms, such as algae and bacteria, or it may be chemical and caused by low pH levels in ocean water. Hereafter we refer to these two dissolution processes simply as net dissolution.."

5. Line 64. Reference Browne et al 2021

R: Done (now 67).

6. Line 80. Add comma after respiration

R: Done (now 85).

7. Line 98. There are some carbonate budget studies that have looked at the impact of acute events on the net CaCO_3 production, which I think are worth mentioning here. Refer to Browne et al 2021 - OMBAR paper which provides a list of carbonate budget studies and what they were used for.

R: Done. Thanks for the recommendation. We have carefully reviewed the study by Browne et al. 2021, and we have cited the works of Perry and Morgan 2017, and Ryan et al. 2019 (now 105-106). All of these studies estimated carbonate budgets after acute impacts.

Furthermore, based on Browne et al. 2021, we have added information to the S2 table

Line 99. I agree - but reading on - does your study address the issue at the reef scape? I dont think so - because its hard to do. Hence we measure a small sub-population and scale up our findings to assess the potential impact at the reefs scape. Obviously the more samples (and coral species) you have - the more confident you can be on your outputs.

R: We agree with this comment and have rephrased the sentence to point out that previous studies have mainly focused on controlled environments or on small spatial scales (line 106).

"...However, measuring net dissolution is difficult and has only been successful in controlled environments or on small spatial scales (Tribollet et al. 2009; Reyes-Nivia 2016; Leggat et al. 2019; Kline et al. 2019), which limits our understanding of the consequences of mass mortality events at the reef scale."

Please also note that the main objective of our study is to quantify the changes in skeletal density following mortality, and we only aimed to provide a first attempt to quantify the potential impacts at reef scales. We now make clear that our estimation is partial, as we only have information for four species (Figure 3), and in the Discussion, we acknowledge the caveats and opportunities further to incorporate fine-resolution measurements of calcium carbonate loss into reef-scape models or carbonate budgets (see lines 224-229).

8. Line 113. I assume this takes into account the percentage of colonies that were affected as well?

R: Yes, our calculations considered colonies affected by SCTLD. Details are provided in the Methods section (lines 511-597) and Figure 3a.

9. Line 141. sup figures are out of order

R: corrected

10. Line 153. yes it seems big when you scale up to the reef - but its the relative amount that is more important. For example, if you're 4 coral colonies only represent 10% of the reef cover - and the rest of the corals are still producing carbonate - then the relative loss might not be that great.

R: Thank you so much for this excellent observation. We now include all species in our models. Please see details in our answer to your comment about Representativeness in the main concerns section.

11. Line 155. I also need to know what proportion of the reef these corals represent and how representative your 10 m² is for a reef that is really quite big! Further given that its the meandriod corals that seem to be suffering the most - a comment on what proportion of the coral cover they represent I also think is important.

R: We have added this information in Figure 3. The abundance of the four coral species relative to the total number of corals (from all species) ranges between 19.01% and 46.81%, depending on the site. And the percentage of CaCO₃ lost attributable to the dissolution of our study species ranges from 9.05% and 80.02%. Please also see our answer to your main comments about 'Representativeness' and 'Sample size' in the main concerns section. We did not only survey 10m² at each reef; we surveyed between 100 and 250 m² in each reef.

12. Line 162. exactly - just these species. What about the rest? Or do these corals represent the lions share of the coral community

R: We now include all species in our models. Please see the previous comment and our answer to your main comment about 'Representativeness'.

13. Line 164. Great to have this to make it comparable to other studies. and its certainly an important loss. But until you know what the total gross carbonate production is for the reef - you dont really know if this will have a significant impact on the reef.

R: Agree, we now include all species in our models. Please see the previous two comments and our answer to your main comment about 'Representativeness'.

Continue...Typically these sub-massive to massive corals grow slowly and dont tend to be the dominant driver of carbonate production on reefs, so just how important is this loss??

R: We recognize that when making carbonate budgets, the potential carbonate contribution of branching species is generally higher than that of submassive and massive species (see Figure 3). Nonetheless, in this study, we are looking at the loss of carbonate in the skeletons.

Because sub-massive and massive species occupy a greater volume than branched species, the loss is also in greater proportion.

14. Line 171 Fig S5

R: corrected

15. Line 171. maybe - but unless you included corals that had died, but not of the disease - can you be 100% certain?

R: We tracked the study site before and during the disease outbreak (see also Molina-Hernández et al. 2022). Therefore, we know the colonies died because of the disease and have a high level of confidence about the date each studied colony died (within 15 days). We add this information in the new version of the MS (lines 379-385).

Text now reads:

*“To measure mass loss in the skeletons after coral death, we collected colony fragments and coral cores from four species: *D. cylindrus*, *P. strigosa*, *S. siderea*, and *O. faveolata*. These species were selected because (i) they are conspicuous species in Puerto Morelos; (ii) they are highly susceptible species to SCTL D (Alvarez-Filip et al 2022); and (iii) we were able to identify (within ~15 days) the date of mortality of the sampled colonies of each species, as we had other ongoing studies in the area (see Cruz-Ortega et al. 2022; Molina-Hernández et al. 2022).”*

16. Line 179. see comment next to figure 2 as well. what comes first - are corals with lower density more likely to contract SCTL D? Looking at your data - this seems a possibility.

R: Many thanks for this observation. The hypothesis that colonies with lower skeletal density were more susceptible to disease due to weakening prior to the SCTL D outbreak seems to be plausible. However, for the species we have information of the same colony before and after mortality (*D. cylindrus*) we also found a strong signal of decreasing skeletal density after mortality, which support our interpretation that skeletal density was reduced by dissolution following mortality. Furthermore, if skeletal density would be associated with the propensity to SCTL D infection, this would be also reflected in *O. faveolata*, as this species is also highly susceptible to the disease. And, for this species we found no significant differences in skeletal density between dead and live colonies.

Please also see our response to your main comment on the Sample design for details of how we have acknowledged this possibility in the Discussion section.

17. Line 193. yes but a lot of the newer studies are using CT scanning which also gives information on changes on the internal structure

R: Due to the extensive restructuring of the discussion section suggested by reviewer 2. This section has been removed. However, we agree CT scan can give more information than x-ray.

Line 203. really need to understand the coral cover here!

R: Thank you. We have added the relative abundance of the four coral species to figure 3. The relative abundance ranges between 19.01% and 46.81%, depending on the site. We now discuss our finding in the context of the total coral community and not just relative to the four studied species (See lines 238-240)

18. Line 259. but as these grow faster - does this offset the potential for faster rates of dissolution?

R: Based on the new analysis where we included the calcification of most species observed during the surveys (see main concerns and methods section lines 404-412), we can state that yes, reef units that have high coverages of key corals in terms of their CaCO₃ production (*Acropora palmata*, *O. faveolata*), buffer the impact caused by the mass loss observed in *D. cylindrus*, *P. strigosa* and *S. siderea* (See Fig. 3).

19. Line 269. see comment next to figure regarding this i.e. does the decline in density with depth a product of declining rates of dissolution, or was the coral already less dense (compared to healthy corals) to begin with and hence was more likely to contract the disease, and therefore suffer continue loss.

R: Please see **Sample design** in the main concerns section.

20. Line 270. but in the healthy colonies - density increased towards the edge.

R: Exactly, the difference in density between live colony samples and dead colony samples is greater at the edge than in deep zones. Therefore, and in agreement with studies that have used experimental blocks in which they observe the same pattern, we suspect that the biomass of microboring organisms is higher at the edge because most of them depend on high light levels.

We have added the following information to be more specific about this (lines 293-296):

"...Higher rates of erosion on the surface area of the colony suggest that the bioerosion was dominated by microborers, most of which are photosynthetic and require a favorable light conditions (e.g., Tribollet and Golubic 2005)."

21. Line 273. ok - so how does that relate to years for the 4 species - so I can see this on the graphs?

R: Recent years correspond to the edge of the core and older years correspond to deep areas of the core. We added this information in the figure 3 to make this relationship more evident.

22. Line 282. I see your point - but looking at table S3 - these same studies have comparable or higher rates of carbonate loss.graphs?

R: True, our rate of 1.32 kg CaCO₃ m⁻² is comparable to results obtained in studies that have used experimental calcium carbonate blocks from corals (please see table S2).

23. Line 295. I worry about this statement for two reasons: you used different coral colonies for 1 year and 2 year analysis, and your sample size is quite low. Combined together (plus the high level of variability in your densities (as seen in figure 2) - you may have randomly picked colonies between the 2 years that have different rates of density loss - so its a product of inherent differences between colonies as opposed to a real effect.

R: To reduce the bias associated with the use of different colonies, the analysis of density changes between the first and second year was only performed on *D. cylindrus* colonies (see Table 1), the species for which the same colonies were sampled before and after mortality. We recognize that the number of replicates for this specific analysis is low and the findings derived from the analysis should be treated with caution. But at the same time, it is consistent with what has been found in different studies (Tribollet and Golubic 2005, Grange et al 2015).

We have rephrased the statement as follows (lines 269-275):

*“We also found evidence that coral skeleton erosion did not occur linearly across time. Following the mortality event of 2018, we resampled the same *D. cylindrus* colonies one and two years later and found that skeletal erosion was not strongly reduced in the second year (Fig. 2A). This is consistent with previous studies indicating that the dissolution rate driven by microbiological activity does not scale over time and instead reaches a plateau a year after substrate exposure (Tribollet and Golubic 2005, Tribollet 2008, Grange et al 2015).”*

24. Line 297. Totally agree that its non-linear, but I thought that rates of bioerosion activity were generally low during the first year and then increased over time due to new bioeroders colonising. Or is this statement true because it just refers to the micro-bioeroders. Clarification please.

R: That's right, in this statement, we only refer to the activity of microborers. We have restructured the statement for clarity (lines 278-280):

“...This pattern can be explained based on the succession of endolithic communities, which is highly dynamic over time (Tribollet and Golubic 2005, Grange et al 2015, Price et al. 2020).”

25. Line 310. surely CT scans of the blocks would address this?

R: The study we cited (Enoch et al. 2021), performed micro-CT analysis before and after deploying the experimental CaCO₃ blocks. The authors observed gains in density due to processes such as cementation being more intense than microerosion.

For clarity, the text now reads as follows (lines 280-289)

“...Contrary to our observations, Enoch et al. (2021) reported gains in substrate density (instead of carbonate dissolution) in experimental substrates deployed on acidified reefs after two years. However, the authors did not rule out that dissolution processes might have preceded cementation (Manzello et al. 2008). While the existing evidence suggests an absence of linearity in dissolution processes (see Tribollet 2008, Grange et al. 2015; Fig 2A), it is still unclear how the succession of dissolution and cementation occurs, and studies are needed to evaluate these processes in different environments (e.g., oligotrophic and acidified sites) with different skeletal types and over different time scales.”

26. Line 316. well depends on the timeframe right?? Surely as the time increases - you get more confident in the rate per unit time.s?

R: Agree, we have restructured the sentence as follows (lines 286-289):

"it is still unclear how the succession of dissolution and cementation occurs, and studies are needed to evaluate these processes in different environments (e.g., oligotrophic and acidified sites) with different skeletal types and over different time scales."

27. Line 321. no idea if its significant because no relative bar to compare it too.

R: We have changed removed the word "significant " to "high" because in this section we just wanted to point out that our reef-level estimates show the total loss of CaCO₃ after a mass mortality event.

28. Line 338. To me, the weakening of the reef structure taken with the reduction in CaCO₃ is the main issue.

R: Fully agree. To emphasize that the weakening of coral skeletons by loss of mass is the main impact on the reef structure, we have restructured the sentence. We address this idea to emphasize its importance. Lines 345-349:

“Overall, our findings highlight the real impact of mass coral mortality for framework loss at ecosystem level and demonstrate vulnerability of Caribbean reefs after a SCTL outbreak. These losses that weaken coral skeletons make colonies more vulnerable to additional threats that may jeopardize the integrity and structural complexity of the reef framework (Fig S5; Eyre et al. 2014, Schönberg et al. 2017).”

Furthermore, according to the suggestion of reviewer 2, this same idea is highlighted in the first lines of the discussion to emphasize its relevance and consequences (lines 229-236).

“...For example, weakened coral skeletons are more vulnerable to fragmentation following tropical storms (Morais et al. 2022) and are likely to be rapidly eroded by macroborers (Molina-Hernandez et al., 2022). In the long term, the net result of the skeletal dissolution rates observed in this study will be a reduction in the structural complexity of the reef framework, which will likely affect the ecosystem services of the coral reefs, such as providing protection and food for commercially important fish species (Fig S4; Eyre et al. 2014, Schönberg et al. 2017).”

29. Line 341. agreed - and there is a movement too!

R: Thank you for the comment. We have also added the study by Browne et al. 2021 (line 362), which highlights the inclusion of microerosion processes in carbonate balances.

30. Line 352. which are?

R: Added to the paragraph (lines 369-374):

*“The Puerto Morelos reef system is located near the northeastern portion of the Yucatan Peninsula in Mexico and is comprised of separate shallow reef units. The reef system covers 9,066 ha and extends over 20 km. The main coral species represented in the shallow reefs are *A. palmata*, *O. faveolata*, *O. annularis*, *P. strigosa*, *S. siderea*, *Agaricia* spp., and *Porites* spp. (Jordan 1979; Caballero-Aragón et al. 2020).”*

31. Line 366. how can cores that are 5 cm long - give you 20+ years of growth?? as shown in figure 2?

R: Most coral cores were at least 15 cm in length. However, some *S. siderea* cores were obtained from 5 cm (n= 5), given that coring this species is challenging due to the density and arrangement of the microstructure. Getting a core from this species can take twice or even three times as long as other species. However, given that *S. siderea* is a slow-growing species (~3.6 mm yr⁻¹), even short core lengths provide information from many years of growth, see table S1.

32. Line 372. and you know this how?

R: Please see our answer to you comment 16.

Text now reads:

*“To measure mass loss in the skeletons after coral death, we collected colony fragments and coral cores from four species: *D. cylindrus*, *P. strigosa*, *S. siderea*, and *O. faveolata*.*

These species were selected because (i) they are conspicuous species in Puerto Morelos; (ii) they are highly susceptible species to SCTL D (Alvarez-Filip et al 2022); and (iii) we were able to identify (within ~15 days) the date of mortality of the sampled colonies of each species, as we had other ongoing studies in the area (see Cruz-Ortega et al. 2022; Molina-Hernández et al. 2022)."

33. Line 422. how were growth years estimated?

R: In order to be more explicit in the methodology employed for the analysis of bands in coral skeletons, we provide more detail in lines (428-438) where we describe the measurements of skeletal density and annual growth):

" Subsequently, the alternating density bands were identified in digitized X-radiographs. The aragonite standard was used to create a calibration curve from which the annual density of each slab ($\text{g CaCO}_3 \text{ cm}^{-3}$) was determined. Annual density series (g cm^{-3}) were obtained from the linear distance along the slabs from the apical zone (recent growth years) to the base of the slabs (old growth years). We used the optical densitometry method described by Carricart-Ganivet and Barnes (2007). The extension rate (cm yr^{-1}) was determined from the distances between density minima peaks. Following this method, we obtained annual density and extension values (one for each pair of bands). The annual density values obtained from the growth bands was considered the units of analysis for this study (see more details in the Statistical Analysis section)."

34. Line 423. how was this determined?

R: Please see our answer to your previous comment.

35. Line 437. there are 7 named on the map?

R: Thanks. Radio Pirata and La Catedral belong to the PM La Catedral reef unit. We have modified the figure legend.

"The sites Radio Pirata and La Catedral are part of the PM La Catedral reef unit"

36. Line 451. I'm confused - I thought this was an equation for volume not mass of C - these are two very different things.

R: We have Corrected this mistake. We change the word mass for volume (line 536).

37. Line 455. so again it should be volume in equation 1

R: Corrected

38. Line. 457. Table S2

R: Done

39. Line 476. so 10 m2 right?

R: No. Each transect was 10 x 1 m, and 10-25 transects were used to survey each reef unit. So, the surveyed area in each reef ranged from 100-250 m². Please see our response to your main comment about the Representativeness.

40. Line 484. Table S1

R: Done

41. Line 488. This assumes that the percentage cover of your 4 corals is consistent over the entire reef area - which is unlikely. Given that you have only surveyed 10 m² per reef, it

would have been good to add in additional broader rapid reef surveys to get a better estimate of the reef community to see if your area is representative. I worry that in selecting an area to sample from - you selected an area that may have had more of these corals to make life easier (which I get!)

R: We agree with your interpretation. However, as mentioned in previous answers, each reef unit was surveyed using at least ten haphazardly allocated band transects of 10 x 1 m (see lines 520-522). So, the total surveyed area was much greater than 10 m² (to 100 from 250 m² in each reef unit).

In addition, please also consider that we surveyed six reef units along the reef system to provide a clearer perspective of the impacts of coral mortality, given that coral community composition was not the same across the entire area. Following some of your main concerns, we now provide the information on the CaCO₃ loss in our four studied species in the context of the carbonate contribution of all other species presented in the site (see Figure 3).

42. Line 671. reference list although covers most of the key lit around dissolution - missing some key bioerosion and more recent important carbonate budget work.

R: Thank you for pointing this out. We appreciate that you have suggested the used of more references to support our findings. We have added to this statement the following recent references: Browne et al. 2021; Hutchings 1986 and 2011; Perry and Morgan 2017; Sokolow 2009; Vega-Thurber et al. 2014, and Molina-Hernandez et al. 2022.

43. FIGURE 1. Is Puerto Morelos the same as La cathedral and radio pirate together - This doesnt match Table s1 as is.

R: Thank you for your comment. We have modified the figure legend to indicate that the sites Radio Pirata and La Catedral are part to the PM La Catedral reef unit (see answer 35).

44. Figure 2. It seems that the colonies that remained healthy and had a high density also had a higher density (with *S. siderea* being the exception) pre the outbreak. To me this suggests that the colonies that contracted the disease were already weakened and therefore that is why they contracted the disease.

R: Please see our response to your main concern regarding Sample design. But briefly, although the hypothesis that colonies with lower skeletal density were more susceptible to disease due to weakening prior to the SCTL D outbreak is plausible, this is not the case in the species that we have samples for before and after mortality from the same colony (*D. cylindrus*), which support our interpretation that skeletal density was reduced by dissolution following mortality in the other two species. Furthermore, if skeletal density would be associated with the propensity to SCTL D infection, this would be also reflected in *O. faveolata*, as this species is also highly susceptible to the disease. And, for this species we found no significant differences in skeletal density between dead and live colonies.

Further, with graphs I to L - the timeframe along the x axis - does this suggest that those corals that died were already on a declining density trajectory pre the outbreak (as per comment above) or that the erosive influence declines as you go deeper into the coral skeleton. If so - the scale should be depth not time as its misleading.

R: The graphs from I to L show the loss of density from the edge to the deeper zones, however, we do not believe that the dead colonies affected by SCTL D had a trajectory of decreasing density before the outbreak, rather this difference between the edge (recent years) and deep zones (old years) is a sign of the intensity of microborers activity. Please see answer 22 for more details.

45. it would also be (more) useful to have the data reported in kg/m²/year - to make it comparable to other studies, which total loss of carbonate is not as its relative to the reef size.

R: Done, we have modified the Y-axis of Figure 3 to kg/m².

46. Figure 3. from these corals, but how does it relate to net carbonate production from the rest of the coral community? Basically I have no idea what percentage of the coral community is comprised of these 4 corals - so yes there is a clear loss, but unless you take in the input from the rest of the coral community (which may then dwarf these numbers) - you don't really get the full impact on the carbonate budget. So yes there is a loss of say 30% on average in total C from these corals, but how does it impact the total CB? Plus you're basing this on 51 colonies over a VERY large area!!!! So knowing the relative impact is important.

R: Please see our response to your main concern regarding the Representativeness of the study. We now provide information for all species surveyed.

47. Table 1. given that the event was in 2018 - how can you be sure that disease was the cause?

R: We had other ongoing studies in the area (Cruz-Ortega et al. 2022; Molina-Hernández et al. 2022). Please see our answer to your comment 32 of how we have described this in the main text.

48. Table 1. so these were different ones that weren't sampled in 2019, but you observed as dead in 2019 hence you can be sure they were dead for 2 years

R: Yes, the colonies of *D. cylindrus* were given special attention because they are rare but conspicuous colonies in the area and were among the first colonies to die completely. Please see our answer to your comment 32.

Reviewer # 2 (Remarks to the Author):

Review for the MS

Coral skeleton dissolution: the other side of mass coral mortality

The MS presents a very interesting topic and attempts an ambitious approach. The data are original and very relevant, and the assessment of naturally established bioeroder communities is commendable. The English language is mostly good.

The MS starts initially strong into the Introduction, however, a major revision including a rewrite of some parts of the MS is required before being publishable. I am also not sure the topic is big enough for the Nature Group. In any case, I think the study design is not particularly strong (low N, unbalanced model, data gaps, very confusing design). The statistics model may be faulty (choice of random and fixed factors, in-/exclusion of factors). The literature could perhaps have been better explored in the bioerosion context.

R: We thank your comments on our manuscript. We have carefully revised your comments and provided a detailed description of how we have amended the manuscript below each comment. Please also note we provide a general description of how we have amended the manuscript in the letter to the editor (above).

We would like to also take the opportunity to say we believe our results will be of interest to the broad readership, as our study offers essential insights into the process that controls the creation and destruction of reef structures. These processes are directly linked to the maintenance and stability of coral reef ecosystems and the goods and services they provide. We show how the processes that occur after coral mortality are essential for understanding ecosystem dynamics in a changing world. In addition, our findings are likely to be representative of many reefs across the world, as mass coral mortality events have unfortunately become increasingly common. Therefore, these findings should be of interest to researchers from a wide range of disciplines and to individuals involved in the management of these reef ecosystems throughout the world.

While the English is good, the content is often confusing or incomplete. The authors did not apply clear wording to explain the bioerosion context, what they did and how they did that. The Methods are incomplete. Parts of the Methods are strewn into the Results and Discussion, and parts of the Results into the Discussion. The statistical model cannot be fully understood from the Methods. The wording for the erosive processes is confusing, and it is not entirely clear whether they only refer to microbioerosion, or to all endolithic bioerosion, or to bioerosion and passive dissolution together? The Discussion is quite bad. It is bloated with information that does not seem to be in context, it does not start strong with pulling the main messages out of their own data, but with topics that skirt around general bioerosion context. I think you can make your MS much stronger and much more believable than you did.

R: We are grateful for the comments of reviewer 2, whose suggestions allowed us to improve the structure of the manuscript. In general terms, we have made changes in each of the sections following this and other specific comments.

Specifically, we have modified some important points in the introduction and methods to make clear that we quantified biological and chemical dissolution through the change in density of living and dead skeletons. In particular we have restructured the discussion following some specific guidelines provided by both referees. Details on how we have amended each section is provided below in the specific comments.

Statistics

The Methods do not adequately explain what you did. I disagree with your choice of what is a random, what is a fixed factor (see my comments in the MS). I am confused about what

and how you calculated, e.g. means. Why did you chop the model into subunits, you increase the risk of errors that way and should initially test as much together as you can. I think you need to include the factors “species” and “reef” in the analyses (anything that does not come out significant can then be dropped in a recalculation to increase the test power), and apparently you tested a few factors that you didn’t properly mention in the Methods, e.g. “penetration depth” (how many levels did you have for the layers in the corals?). I think you should clearly and consistently replace “year” with “exposure time” (0 years, 1 year, 2 years). I do not understand how you can conduct parametric tests like an ANOVA with your low sample sizes and sample sizes under, you would never get normal-distributed data, and you have an incredibly high risk for errors. But there seem to be more data points in the figures than replicates in the table, please explain. I also do not understand why you did not use a balanced model with the same sample size in every group, because variable group sizes have a strong effect on the variation with such small samples. So, all this creates reasonable doubt about your data, on top of all the confusion what was really done.

R: Thank you for your comments. Many of your concerns are related to lack of detail we provided in the first version of the manuscript. We made substantial changes to this section to give a better description of our rationale and the justification of the criteria and model selection. Our statistical approach is described in lines 471-509. In addition, below we provide a brief justification to each of the points raised in this comment:

- **Sample size.** This point was also raised by the first referee. As mentioned above, the problem relies in our lack of detail in the description of the sampling design. We have now improved the description in lines 471-491.

Briefly, the observational units used to construct the statistical models, are the annual density data obtained from the multiple annual bands of coral growth (not the individual coral colonies). For this reason, in Figure 2 in plots A-D, the data exceed the number of cores listed in Table 1. We now provide, in the supplementary methods (appendix 2; table S1), a list of the annual data observed in each coral core, i.e., all our observation units.

These annual density estimates were obtained from optical densitometry of each of the cores. On average, 14 annual estimates were obtained for each core (see supplementary information). The total sample size include in our models is therefore 654 (see dots in Figure <2 A-D). Using this approach not only increases the number of observations but, since it is based on the analysis of bands along the cores (from the edge of the colony to the center of the colony), it gives a clear perspective of the density change in different sections within the skeletons.

We would like to also note that this hierarchical structure of the data was considered in the construction of the linear mixed models. In addition, in this new version of the MS, we provide a detailed explanation of all this information in the methods section (lines 479-491).

Balanced design. A balanced design could not be followed given two main reason: first, the natural availability of species (e.g., *D. cylindrus*) restricted the number of samples we could obtain. But also, as mentioned above our sample size were not the individual coral fragments, but the annual density bands from each colony. Given that species have different skeletal density and growth rates, each fragment represent a different number of bands (i.e, years). However, as explained above, our sample size is large enough to account for the variability associated to unbalanced model. Also, please see below our answer to the structure of the model.

- **Use of separate models for each species.** We decided to use a different model for each species because: a) each species has different skeletal microstructures, sizes, and morphologies, and more importantly, each core of each species has different initial skeletal densities (mean density of live colonies, Fig. 1 a-e). b) one species (*D. cylindrus*) has a different sampling design. For *D. cylindrus* the same colonies were sampled before and after mortality; and samples were collected in three times (before mortality, after 1 year and after 2 years of mortality). While for the other three species, different colonies were used for only two periods (before and 2 years after mortality). All these characteristics mean the corals species (substrates) can be considered independent observations.

In addition, while a general model (grouping species together), in principle, might appear to be a more robust model due to having more levels in each variable. Creating species-independent models offers more insights to explain the differences between live and dead coral skeletal density because the effect of the variation that might exist between species (rather than blocking the variables in a general model) is reduced to a minimum (Castillo 2011).

However, despite the above we understand is possible to include all species in the same model. We thus fitted two alternative models. One that contains data for all four species (i.e., including *D. cylindrus*), and one with just data from the three species for which the sampling method was consistent (see above). Overall, the results of both models show the same significant differences between dead and alive *D. cylindrus*, *P. strigosa* and *S. siderea* observed in the individual models (see appendix 2 General Model diagnostic).

Fixed and random effects. We selected as fixed factors the state of the colony (alive or dead) because our main objective was precisely to observe changes in the density of the skeletons after mortality. We also used the year in which the density value was obtained, that is, the location or section of the annual growth band within the coral skeleton, to test for differences in dissolution rate along the skeleton (from the border section to the deep sections). We have made this clear in lines 479-491.

Random factors (colony identity, reef zone, site) were chosen because recognize these variables can exert an effect on the variability of density in both living and dead skeletons, but this variation is not within our research question.

Methods

The Methods section is inadequate and quite confusing. Did you subsample corals where you had N=1 (that would be another random factor to test)?

R: We have added details in the methods section to make clearer the sampling design and replicates obtained in each species. Please see lines 387-402.

Briefly, the only case where we obtained an N= 1 was in the colonies of *D. cylindrus* that inhabited La Pared and Radio Pirata reefs. when alive with no signs of diseases in 2015, then one year after the total mortality (2019) and two years after total mortality (2020)

As the reviewer points out, because subsamples were obtained at different times in the three *D. cylindrus* colonies, colony identity (ID) was considered as a random factor in the statistical model (LMM). Furthermore, in this model we allowed the intercept and slope to be fitted to each ID. See supplementary material (please see appendix 2).

Did you look into your samples to see what bioeroders you had? It sounds as if you excluded macroborers from the analysis, did you do this with the thresholding? Can you make that clearer?

R: We thank your comment as it gave us the opportunity to make clear in the manuscript that we only focus on changes in skeletal density attributable to microerosion or dissolution. This is now clearly mentioned in the introduction (lines 112-115):

“In this study, we measured density changes produced by net dissolution to quantify CaCO₃ mass loss in the dead skeletons of four reef-building corals after an outbreak of stony coral tissue loss disease (SCTLD) in the Puerto Morelos reef system (Alvarez-Filip et al. 2022).”*

In lines 69-70 we refer that the net dissolution is the process of microborring and chemical dissolution together

And methods (lines 465-469):

“In this study, we were only interested in measuring density changes in coral skeletons resulting from net erosion (microerosion and dissolution processes); therefore, we only measured sections of the coral cores that had not been eroded by internal macroborers (e.g., bivalves, and worms).”

I think you need to explain the importance of your 4 chosen corals for the ecosystem in general

R: Thanks for this suggestion. We have included information on the importance of these 4 species in lines 379-385.

*“...To measure mass loss in the skeletons after coral death, we collected colony fragments and coral cores from four species: *D. cylindrus*, *P. strigosa*, *S. siderea*, and *O. faveolata*. These species were selected because (i) they are conspicuous species in Puerto Morelos; (ii) they are highly susceptible species to SCTLD (Alvarez-Filip et al 2022); and (iii) we were able to identify (within ~15 days) the date of mortality of the sampled colonies of each species, as we had other ongoing studies in the area (see Cruz-Ortega et al. 2022; Molina-Hernández et al. 2022).”*

In addition, we have made a significant effort to analyze the total CaCO₃ contribution of all species observed in our surveys. We are confident that this new approach improves the perspective of the representativeness of the dissolution observed in the skeletons of these four species in the context of the CaCO₃ contribution of the rest of the species.

And I think you should somehow state that even though some corals were sampled multiple times (which needs to be reflected in the stats, these samples are not independent), and sometimes different corals were sampled at the same time for live/dead comparisons, you should state that the focus is on time of exposure to bioerosion, so the effect for the variation in your calculations is still more or less the same. I am not sure, but I think you divided the calculations depending on that sampling strategy? Anyway, I did not fully understand what was done, which is not good. And all topics need to be introduced in the Methods, they cannot suddenly occur in the Discussion without prior mentioning.

R: Thank you very much for this great observation. As you, we think that because all the analyzed colonies of all the species died almost at the same date (<15 days), the time of exposure of the skeletons to the inclemency of micro-erosive processes was the same. The

only expression was the species *D. cylindrus* that has fragments collected at 1 and 2 years after the death of the colonies. For this reason and other details (see response Use of separate models for each species in Statistics) due to the specific characteristics of each species (as detailed above), we split the analyses.

We have modified different sections of the paper to address this concern and give clarity to each process used.

I don't understand why *O. faveolata* was included in the study, but in the data analysis it was not used like the other coral species? Was there a reason? Can you not still include these data?

R: Thank you for this comment. *O. faveolata* was included in the study because of its importance in terms of CaCO₃ fixed in its skeleton, being a species highly affected by SCTLTD and being well represented in all reef units (lines 379-385). However, this was the only species in which no significant change in skeletal density was observed between live and dead colonies (see Figure 1e and 2D). For this reason, initially we did not consider it for the reefscape analysis. However, following several suggestions of the first referee, we now include all the species presented in our study sites (including *O. faveolata*) and not only the three for which we have robust evidence of declines in skeletal density (*D. cylindrus*, *P. strigosa* and *S. siderea*). For the updated version of the analysis, we used the same amount of CaCO₃ for *O. faveolata* for before and after the mortality (as we did not find evidence of change for this species). Please see Figure 3, and lines 167-191 in the results section:

*“According to our estimations, the widespread mortality of coral colonies led to an enormous reduction in fixed CaCO₃ at the reef scale (Fig. 3). The standardized erosion rates are equivalent to area-wide total net losses of CaCO₃ ranging between -140 and -1,471 t CaCO₃ km⁻², depending on the reef site, which can be exclusively attributed to the dissolution of the dead skeletons of *D. cylindrus*, *P. strigosa*, and *S. siderea* (Fig. 3 a-c).”*

Discussion

The Discussion needs a rewrite. I think the present Discussion is damaging to your paper and that a rewrite will hugely improve the MS. You need to start in with your own data and then spread out into marginal topics that you may glean from the literature. What is your strongest point? Probably the substrate loss at reef level, so maybe start with that. Your big bang here is the % loss at ecosystem level, which means that the whole community is changed. This is something loud and strong to start in with. Then you go from there and explain what mattered most in your choice of factors, and you explain the differences of the corallites etc., but you need to come back and let the readers know which coral species was dominant etc. so that we understand what this does to the reef community. From there you can go into more microborer-related stuff such as the plateauing and the preferred penetration level etc. I disagree with a few statements in the Discussion. You need to more strongly consider that coral microborers are mostly photosynthetic and need light. And that expansion slows down when crowding in the favored light horizon occurs.

R: We very much appreciate this comment. We have substantially modified the discussion following your suggestions. The first paragraphs now address the consequences of the percentage of CaCO₃ loss.

“Our results reveal significant losses in the mass of dead coral colonies following skeletal exposure and provide quantitative insights into how living coral tissues prevent losses of the reef matrix. Tissue mortality caused by SCTLTD resulted in changes in the composition of the microendolithic community from one that can positively interact with the coral to one that is

predominately eroded (Fig. S3; Le Campion-Alsumard 1995; Fine and Loya 2002). Thus, the loss of CaCO₃ observed in the dead coral skeletons in this study was due to a heightened increase in net dissolution following the loss of protective tissue cover, microperforation, and the metabolic activity of the epilithic and endolithic algal and bacterial communities that colonized the skeletons (Leggat et al. 2019). The lack of protection by living tissues, coupled with the naturally high surface area of scleractinian coral skeletons, created conditions that were favorable to CaCO₃ dissolution (Ricci et al. 2019). This process can be classified as a type of succession. Initially, exposed skeletons are almost exclusively dominated by microborers that modify the substrate, making it accessible to macroborers and grazing epilithic bioeroders, which become increasingly important over time (e.g., Chazottes et al. 1995; Pari et al. 2002; Tribollet and Golubic 2005; Grange et al. 2015). In turn, the bioeroders increase the surface area of the substrate by creating internal cavities and removing alternative surface covers, which creates conditions that favor passive dissolution (Chazottes et al. 1995; Tribollet and Golic 2005).

Skeletal dissolution following mass mortality events threatens the structure and long-term stability of the reef matrix. The CaCO₃ losses attributed to the widespread mortality of the colonies of only three coral species represented a reduction of almost 7% of the total amount of CaCO₃ fixed in the carbonate skeletons of the living corals of all species in our study system (Fig. 3d). This loss, which was due to skeletal dissolution and occurred in less than two years, is substantial, considering that the accumulation of CaCO₃ in these skeletons occurred over tens or even hundreds of years. Our findings highlight the impact of widespread coral die-off in terms of CaCO₃ loss over large geographical areas and bring the potential implications of SCTL outbreaks at the regional level into perspective. However, it is crucial to consider that our findings only reflect the first of many destructive processes that follow coral mortality (Grange et al. 2015). Therefore, we can expect that total CaCO₃ loss over the long-term will be notably higher than what we have quantified. This is particularly important given that mass loss makes coral skeletons more susceptible to other erosive and destructive forces. For example, weakened coral skeletons are more vulnerable to fragmentation following tropical storms (Morais et al. 2022) and are likely to be rapidly eroded by macroborers (Molina-Hernandez et al., 2022). In the long term, the net result of the skeletal dissolution rates observed in this study will be a reduction in the structural complexity of the reef framework, which will likely affect the ecosystem services of the coral reefs, such as providing protection and food for commercially important fish species (Fig S4; Eyre et al. 2014, Schönberg et al. 2017).”

In addition, we have made several other changes to the discussion following your comments.

Rates

You very much need a time reference in your rates (most used is: kg bioerosion per m² and year, but any unit is OK, as long as it is there, it can be re-calculated for comparisons). Bioerosion rates only become comparable to other studies when they have a reference to time. Of course: There usually is a higher rate in a year than in a month.

R: Thank you for your comment. Following this suggestion, we have added a microerosion/dissolution rate to make our results comparable with other studies.

In the results section in lines 173-178, we make clear that:

“...The total amount of CaCO₃ lost across the shallow habitat of the entire reef system (~3,145,967.15 m²) was equivalent to -4184 t, which represents an average dissolution of -1.33 kg CaCO₃ m⁻².”

Please also note that we have updated figure 3 b-c to represent the y-axis from tons to kg. So it is in the same units as in the rest of the studies.

I hope you will consider my comments, even though they will cause you much work. Even though I have not fully understood your data structure, I believe your MS is publishable, and that you can make it significantly stronger than it is. Within-MS comments as below.

R: We thank you for the comments to the manuscript. The feedback was very helpful and gave us the opportunity to substantially improve the manuscript. We hope you will consider that our article is now suitable for publication.

Minor comments (changes made the Referee directly in the text)

1. Line 10 Text changed to “...fixed within coral skeletons through erosion,..” words “and dissolution” removed.
2. Lines 17-18 The text has been modified according to the suggestion.

Text now reads:

“...which suggests that the metabolic activity of these organisms caused or contributed to the dissolution of dead coral skeletons.”

3. Line 35 Add reference “Heileman and Mahon 2009”
4. Lines 35-38 The text has been modified according to the suggestion.

Text now reads:

“...The three-dimensional structure of each coral reef is a product of CaCO₃ production and erosion stemming from the environment and a multitude of organisms, with these processes acting across various spatiotemporal scales”

5. Line 48 Text changed to “...This transition is even more concerning when modelled scenarios predict...” Changed “future” to “modelled”.
6. Lines 74-75 Text changed to “...Second, existing microbial endolithic communities...” Add “microbial”.
7. Line 77 Add reference “Fine et al. 2006”
8. Lines 87-88 Text changed to “...The interstitial water that penetrates the pores of coral skeletons...” Add “coral”.
9. Line 90 Add reference “Risk and Muller 1983”

10. Lines 93-94 Text changed to "...CaCO₃ dissolution is of particular concerning, ..."
Changed "Carbonate" to "CaCO₃".

11. Lines 100-102 Text changed to "Internal macrobioerosion and the net dissolution of coral skeletons are particularly concerning during mass mortality events, such as those due to widespread bleaching or disease outbreaks that result in the loss of the protective cover of coral tissue". Add "that results in the loss of the protective cover of coral tissue".

12. Line 119 Reviewer #2's suggestion made in this line "These four coral species were chosen because" is addressed in the material and methods section (lines 381-385).

Text now reads:

"...These species were selected because (i) they are conspicuous species in Puerto Morelos; (ii) they are highly susceptible species to SCTL D (Alvarez-Filip et al 2022); and (iii) we were able to identify (within ~15 days) the date of mortality of the sampled colonies of each species, as we had other ongoing studies in the area (see Cruz-Ortega et al. 2022; Molina-Hernández et al. 2022)."

13. Line 127 Text changed to "... (Fig. 1a)..." Add "a".

14. Lines 127-129 Text changed to "For *D. cylindrus*, we obtained core samples from colonies in 2015, when they were alive, and then sampled the same colonies in 2019 and 2020, one and two years after they had died in 2018 (Table 1A)."
Changed "and from the same colonies in 2019 and 2020 after they had already died" to "and two years after they had died in 2018 (Table 1A)."

15. Lines 129-131 The text has been modified according to the suggestion.

Text now reads:

"...For *S. siderea*, *P. strigosa*, and *O. faveolata* we obtained samples during the same sampling campaign (in 2020) from live colonies and from colonies that died due to the SCTL D outbreak in 2019 (see methods; Table 1B)."

16. Lines 139-156 The text has been modified according to the suggestion.

Text now reads:

"...We found strong evidence of density loss in carbonate structures after coral tissue mortality for three out of the four studied species (Fig. 2). Separate LMM's showed skeletal density was significantly lower after mortality lower in *D. cylindrus* ($\chi^2 = 14.071$, $df = 2$, $p < 0.001$), *P. strigosa* ($\chi^2 = 10.861$, $df = 1$, $p < 0.001$), and *S. siderea* ($\chi^2 = 6.021$, $df = 1$, $p = 0.014$). While, the model for *O. faveolata* we did not observe significant differences between the skeletal density of dead and alive colonies ($\chi^2 = 0.009$, $df = 1$, $p = 0.921$; Fig. 2 H). The porosity analyses supported the findings obtained by optical densitometry. Higher porosity was observed in

dead *D. cylindrus*, *S. siderea*, and *P. strigosa* colonies compared to live colonies (Fig. S1).

For *D. cylindrus* we were also able to explore the progression of skeletal density loss using data from the two sample periods after mortality. We found significant differences in the colony density before and after the colonies died (1 year after death, $z = 10.798$, $p < 0.001$; 2 years after death, $z = 6.586$, $p < 0.001$). However, no differences in density were found between colonies one year after death and two years after death ($z = -2.27$, $p < 0.062$; Fig. 2 E). This suggests that the most pronounced change occurred during the first year following mortality.”

17. Lines 166-176 The text has been modified according to the suggestion.

Text now reads (168-178):

“...According to our estimations, the widespread mortality of coral colonies led to an enormous reduction in fixed CaCO_3 at the reef scale (Fig. 3). The standardized erosion rates are equivalent to area-wide total net losses of CaCO_3 ranging between -140 and -1,471 t $\text{CaCO}_3 \text{ km}^2$, depending on the reef site, which can be exclusively attributed to the dissolution of the dead skeletons of *D. cylindrus*, *P. strigosa*, and *S. siderea* (Fig. 3 a-c). The total amount of CaCO_3 lost across the shallow habitat of the entire reef system ($\sim 3,145,967.15 \text{ m}^2$) was equivalent to -4184 t, which represents an average dissolution of -1.33 kg $\text{CaCO}_3 \text{ m}^2$. The CaCO_3 losses attributable to these three species represent a reduction of 6.78% of the total CaCO_3 fixed by all scleractinian coral colonies in the entire reef system after one single mortality event (Fig 3d).”

18. Lines 365-410. Reviewer #2 provided several modifications to the site and sample collection section, these suggestions together with the recommendations of reviewer #1 have resulted in significant changes to this section.

Text now reads (lines 379-412):

“To measure mass loss in the skeletons after coral death, we collected colony fragments and coral cores from four species: *D. cylindrus*, *P. strigosa*, *S. siderea*, and *O. faveolata*. These species were selected because (i) they are conspicuous species in Puerto Morelos; (ii) they are highly susceptible species to SCTLTD (Alvarez-Filip et al 2022); and (iii) we were able to identify (within ~ 15 days) the date of mortality of the sampled colonies of each species, as we had other ongoing studies in the area (see Cruz-Ortega et al. 2022; Molina-Hernández et al. 2022).

We used two different sampling designs, as previously collected samples of *D. cylindrus* were available. In 2015, *D. cylindrus* colonies that were living with no signs of disease were identified and sampled (Cruz-Ortega et al. 2022). In this study, we repurposed and reprocessed these previous samples and sampled the same colonies two additional times: one year after the total mortality event (2019) and two years after the mortality event (2020; Table 1; Figure 1a). For the other three species (*P. strigosa*, *S. siderea*, and *O. faveolata*), we sampled living and dead colonies in 2020 (Table 1; Figure 1a). All cores from dead colonies were obtained from colonies that were known to have died due to SCTLTD, and the date of mortality was recorded (within 15 days). In this case, all samples from dead colonies reflected two years of exposure. The samples were obtained from different colonies found within an area of $\sim 30 \text{ m}^2$ in La Catedral (Figure 1a). Samples from

all colonies (living and dead) were collected from the apical part of the colony. Fragments obtained from *D. cylindrus* colonies were ~ 10 cm in diameter and 10–20 cm in length (Table 1A). Cores obtained for *P. strigosa*, *S. siderea*, and *O. faveolata* were 3 cm in diameter and 5–15 cm in length. Data for all colony fragments and coral cores are shown in Table 1B.

To contextualize the possible mass loss of *D. cylindrus*, *S. siderea*, *P. strigosa*, and *O. faveolata* with the CaCO_3 contributed by the coral community, we obtained fragments and cores of the most representative scleractinian species within the Puerto Morelos reef system. All samples were collected at La Catedral (see complementary methods; *Acropora palmata* = 2, *Undaria agaricites* = 4, *Undaria teunifolia* = 4, *Orbicella annularis* = 4, *Porites astroides* = 8, *Pseudodiploria clivosa* = 3, *Porites porites* = 4, *Stephanocoenia intercepta* = 2, *Montastrea carvernosa* = 4, *Dichocoenia stokessi* = 6, *Colpophyllia natans* = 3, and *Siderestrea radians* = 4). For all samples of these species, skeletal density was calculated as described below.”

19. Line 420. Changed “Hungary” to “healthcare”
20. Lines 420-421. Text changed to “...The slabs were oriented longitudinally in rows for radiographic scanning (2–3 slabs per row), alternating dead and live slabs...” Add “radiographic scanning”.
21. Line 460-462 Text changed to “...The porosity analysis offered information about air spaces according to the spatial resolution of all parts of the slabs...” Add “about”.
22. Lines 465-469 The text has been modified according to the suggestion “Macrobioerosion was excluded from this analysis by...”

Text now reads:

“...In this study, we were only interested in measuring density changes in coral skeletons resulting from net erosion (microerosion and dissolution processes); therefore, we only measured sections of the coral cores that had not been eroded by internal macroborers (e.g., bivalves, and worms).”

23. Line 512. Text changed to “...We estimated the loss of CaCO_3 associated ...” Add “estimated”.
24. Line 513-514. Text changed to “...colonies due to SCTL D outbreak in the Puerto Morelos reef system...” Add “outbreak”.
25. Line 520. Text changed to “...from six reef sites across the reef system...” Add “from”.
26. Lines 536-537. Text changed to “...and r^2 is the mean colony diameter...” Add “the”.
27. Lines 586. Text changed to “...Lastly, we used the above estimates of CaCO_3 loss to predict ...” Changed “carbonate” to “ CaCO_3 ”.
28. Lines 592-594. Text changed to “...To determine the loss of CaCO_3 in the entire Puerto Morelos reef system, the amount of CaCO_3 contained in each reef was

*calculated for each period (before and after the SCTL D outbreak)". Add "entire"
"contained" and "calculated".*

REVIEWERS' COMMENTS:

Reviewer #1 (Remarks to the Author):

The authors have provided a thorough re-write and response to my previous comments and concerns. In addition, I have also carefully read reviewer 2's concerns, which largely focused on the methods, statistics and discussion. Key points to note:

1. I was pleased to see that the authors had provided more details and analysis around the coral reef community to place their data into clearer context.
2. Following both my and reviewer 2's comments on the sample size, the authors provided a much more detailed description of their methods along with justification of their statistical model selection. Here they have highlighted that the observational units are the density estimates and not the cores (which are the sampling units), and as such the number of samples is higher.
3. Further, the model choice to me makes sense as does the selection of fixed and random factors - which again has been further explained and backed up by previous papers that have used a similar approach.
4. Reviewer 2 also highlighted that the design was unbalanced which is a greater issue with small sample sizes, however using the density estimates as the samples of which there are more, then this is less of an issue.
5. Reviewer 2 also suggested a model which included all coral species. I would disagree with that - given the arguing points highlighted by the authors - but kudos to them, they did also try this and found no big difference in the outcome.
6. Reviewer 2 also had some concerns with the discussion and provided some good pointers to improve. I can see that these have been taken into account - although I do have one suggestion here for further improvements (see below)

Points for further improvement:

1. the first paragraph of the discussion is not great. It starts well but then gets into the weeds of literature rather than focusing on the study. The second paragraph is much stronger. I would suggest that the authors remove line 196 to 213 to further down in the discussion - with a paragraph focusing on the detailed process of dissolution, and then merge line 194 to 196 to the paragraph starting on line 215 - and have this as your opening paragraph.
2. in the abstract it states that there is a reduction in caco3 sequestered from 23.9% to 33.9% (shouldn't it be the other way around??). Nowhere else are these numbers mentioned, instead the authors focus on the 7% decline in caco3 sequestered - which makes more sense to me. I would stick with that for consistency.

Aside from these minor changes, I'm happy for the manuscript to be published.

REVIEWERS' COMMENTS:

Reviewer #1 (Remarks to the Author):

The authors have provided a thorough re-write and response to my previous comments and concerns. In addition, I have also carefully read reviewer 2's concerns, which largely focused on the methods, statistics and discussion. Key points to note:

1. I was pleased to see that the authors had provided more details and analysis around the coral reef community to place their data into clearer context.
2. Following both my and reviewer 2's comments on the sample size, the authors provided a much more detailed description of their methods along with justification of their statistical model selection. Here they have highlighted that the observational units are the density estimates and not the cores (which are the sampling units), and as such the number of samples is higher.
3. Further, the model choice to me makes sense as does the selection of fixed and random factors - which again has been further explained and backed up by previous papers that have used a similar approach.
4. Reviewer 2 also highlighted that the design was unbalanced which is a greater issue with small sample sizes, however using the density estimates as the samples of which there are more, then this is less of an issue.
5. Reviewer 2 also suggested a model which included all coral species. I would disagree with that - given the arguing points highlighted by the authors - but kudos to them, they did also try this and found no big difference in the outcome.
6. Reviewer 2 also had some concerns with the discussion and provided some good pointers to improve. I can see that these have been taken into account - although I do have one suggestion here for further improvements (see below)

R: We thank your positive assessment of the new version of our manuscript. We have carefully revised your comments and provided a detailed description of how we have amended the manuscript below each comment.

Points for further improvement:

1. the first paragraph of the discussion is not great. It starts well but then gets into the weeds of literature rather than focusing on the study. The second paragraph is much stronger. I would suggest that the authors remove line 196 to 213 to further down in the discussion - with a paragraph focusing on the detailed process of dissolution, and then merge line 194 to 196 to the paragraph starting on line 215 - and have this as your opening paragraph.

R: Thank you. Based on the valuable suggestion of Reviewer #1, we have reworded the first two paragraphs of the Discussion section. We agree that this is a better start for the Discussion and appreciate the recommendation.

2. in the abstract it states that there is a reduction in CaCO_3 sequestered from 23.9% to 33.9% (shouldn't it be the other way around??). Nowhere else are these numbers mentioned, instead the authors focus on the 7% decline in CaCO_3 sequestered - which makes more sense to me. I would stick with that for consistency.

R: We agree with this observation. We have modified this line and now refer to a 7% decline in CaCO₃, which is consistent with the information we provide in the manuscript.

Aside from these minor changes, I'm happy for the manuscript to be published.